# Phytosome-Encapsulated 6-Gingerol- and 6-Shogaol-Enriched Extracts from *Zingiber officinale* Roscoe Protect Against Oxidative Stress-Induced Neurotoxicity

**DOI:** 10.3390/molecules29246046

**Published:** 2024-12-22

**Authors:** Nootchanat Mairuae, Parinya Noisa, Nut Palachai

**Affiliations:** 1Biomedical Research Unit, Faculty of Medicine, Mahasarakham University, Mahasarakham 44000, Thailand; nootchanat.m@msu.ac.th; 2School of Biotechnology, Institute of Agricultural Technology, Suranaree University of Technology, Nakhon Ratchasima 30000, Thailand; p.noisa@sut.ac.th

**Keywords:** apoptosis, ginger, neuroinflammation, oxidative stress, phytosome, 6-gingerol, 6-shogaol, *Zingiber officinale* Roscoe

## Abstract

The rising prevalence of neurodegenerative disorders underscores the urgent need for effective interventions to prevent neuronal cell death. This study evaluates the neuroprotective potential of phytosome-encapsulated 6-gingerol- and 6-shogaol-enriched extracts from *Zingiber officinale* Roscoe (6GS), bioactive compounds renowned for their antioxidant and anti-inflammatory properties. The novel phytosome encapsulation technology employed enhances the bioavailability and stability of these compounds, offering superior therapeutic potential compared to conventional formulations. Additionally, the study investigates the role of the phosphoinositide 3-kinase (PI3K)/protein kinase B (Akt)-signaling pathway, a key mediator of the neuroprotective effects of 6GS. Neurotoxicity was induced in SH-SY5Y cells (a human neuroblastoma cell line) using 200 μM of hydrogen peroxide (H_2_O_2_), following pretreatment with 6GS at concentrations of 15.625 and 31.25 μg/mL. Cell viability was assessed via the MTT assay alongside evaluations of reactive oxygen species (ROS), antioxidant enzyme activities (superoxide dismutase [SOD], catalase [CAT], glutathione peroxidase [GSH-Px]), oxidative stress markers (malondialdehyde [MDA]), and molecular mechanisms involving the PI3K/Akt pathway, apoptotic factors (B-cell lymphoma-2 [Bcl-2] and caspase-3), and inflammatory markers (tumor necrosis factor-alpha [TNF-α]). The results demonstrated that 6GS significantly improved cell viability, reduced ROS, MDA, TNF-α, and caspase-3 levels, and enhanced antioxidant enzyme activities. Furthermore, 6GS treatment upregulated PI3K, Akt, and Bcl-2 expression while suppressing caspase-3 activation. Activation of the PI3K/Akt pathway by 6GS led to phosphorylated Akt-mediated upregulation of Bcl-2, promoting neuronal survival and attenuating oxidative stress and inflammation. Moreover, Bcl-2 inhibited ROS generation, further mitigating neurotoxicity. These findings suggest that phytosome encapsulation enhances the bioavailability of 6GS, which through activation of the PI3K/Akt pathway, exhibits significant neuroprotective properties. Incorporating these compounds into functional foods or dietary supplements could offer a promising strategy for addressing oxidative stress and neuroinflammation associated with neurodegenerative diseases.

## 1. Introduction

In recent years, the global prevalence of neurodegenerative conditions has increased, raising significant health concerns. At the core of these concerns lies the progressive loss of neurons, driven by apoptosis, oxidative stress, and neuroinflammation [1,2]. This neuronal cell death is a critical factor in the gradual decline observed in diseases such as Alzheimer’s, Parkinson’s, Huntington’s, and Amyotrophic Lateral Sclerosis (ALS). For instance, in Alzheimer’s disease, the buildup of beta-amyloid plaques and tau protein tangles initiate apoptotic pathways, characterized by mitochondrial dysfunction, heightened oxidative stress, and inflammatory responses [3,4]. Parkinson’s disease similarly involves the deterioration of dopaminergic neurons, propelled by the aggregation of alpha-synuclein protein, leading to mitochondrial dysfunction, oxidative stress, and inflammation [5,6]. Huntington’s disease is marked by the accumulation of mutant Huntingtin protein, disrupting cellular functions and calcium balance, further intensifying oxidative stress-induced neuronal apoptosis [7,8]. Likewise, ALS, often linked to mutations in genes like SOD1 and TARDBP, triggers motor neuron death through mitochondrial dysfunction, elevated oxidative stress, and neuroinflammation [9,10].

Despite advances in understanding these diseases, current treatments focus mainly on symptom relief and slowing disease progression but do not effectively address the root causes [11]. Consequently, there is an urgent need to explore preventive approaches that target the underlying mechanisms of neurodegeneration, with the goal of developing more effective therapies.

Furthermore, 6-gingerol and 6-shogaol, bioactive compounds derived from *Zingiber officinale* Roscoe (ginger), exhibit significant neuroprotective effects by inhibiting apoptosis, reducing oxidative stress, and modulating neuroinflammation, processes closely linked to neuronal loss in neurodegenerative diseases [12,13]. Although these compounds share neuroprotective properties, they differ structurally: 6-gingerol is abundant in fresh ginger, while 6-shogaol forms during the drying or cooking process. Both compounds have been shown to scavenge free radicals and upregulate endogenous antioxidant enzymes like SOD, GSH-Px, and CAT, protecting neurons from oxidative stress [14,15]. Additionally, they exert anti-inflammatory effects by suppressing microglial activation and the production of pro-inflammatory cytokines such as interleukin-1β and TNF-α [16,17]. Preclinical studies in animal models of Alzheimer’s and Parkinson’s diseases have demonstrated their potential to improve cognitive function, alleviate motor deficits, and reduce neuronal loss, highlighting their promise as therapeutic agents [18,19].

However, one of the main limitations of using 6-gingerol and 6-shogaol as therapeutic agents is their poor bioavailability. These compounds undergo rapid metabolism and limited absorption, reducing their overall efficacy when used in conventional forms [20]. To overcome this challenge, phytosome encapsulation technology has emerged as a promising solution. By encapsulating bioactive compounds within a phospholipid-based carrier system, phytosomes enhance the absorption and bioavailability of these compounds, facilitating better cellular uptake and prolonged therapeutic effects [21,22]. This technology is particularly beneficial for compounds like 6-gingerol and 6-shogaol, where enhanced bioavailability could amplify their neuroprotective actions against oxidative stress and inflammation in neurodegenerative conditions.

While 6-gingerol and 6-shogaol have shown promise in modulating oxidative stress and inflammation, their neuroprotective mechanisms in H_2_O_2_-induced neurotoxicity in SH-SY5Y cells remain unclear and have not been fully explored. Therefore, this study investigates the neuroprotective potential of phytosome-encapsulated 6-gingerol- and 6-shogaol-enriched extracts from *Zingiber officinale* Roscoe against H_2_O_2_-induced cytotoxicity in SH-SY5Y cells. To gain insights into the underlying mechanisms, we assessed various parameters, including the activities of antioxidant enzymes such as CAT, SOD, and GSH-Px, levels of MDA and ROS, and the expression of key apoptotic factors (Bcl-2 and caspase-3), the PI3K/Akt pathway, and inflammation markers like TNF-α.

## 2. Results

### 2.1. Identification of Active Compounds and Their Biological Activities

The analysis in Table 1 reveals the presence of active compounds in the aqueous extract of *Zingiber officinale* Roscoe, with total phenolic content and total flavonoids measured at 207.60 ± 2.25 mg/g and 25.22 ± 0.48 mg/g of the extract, respectively. The 50% hydroalcoholic extract demonstrated higher concentrations of these bioactive compounds, with total phenolics reaching 315.43 ± 3.01 mg/g and flavonoids at 68.11 ± 0.48 mg/g. The 95% hydroalcoholic extract also contained significant levels of these compounds at 251.48 ± 0.42 mg/g for phenolics and 47.72 ± 0.08 mg/g for flavonoids. These results indicate that the 50% hydroalcoholic extraction yields the highest concentration of phenolic and flavonoid compounds, with a statistically significant difference (*p* < 0.001) when compared to the aqueous and 95% hydroalcoholic extracts.

To assess the therapeutic potential of the 50% hydroalcoholic extract, further tests were conducted to evaluate its antioxidant and anti-inflammatory properties, which are particularly relevant to neurodegenerative conditions. As shown in Table 1, the 50% hydroalcoholic extract exhibited significantly lower EC50 values in antioxidant assays (DPPH, FRAP, and ABTS) and COX-2 inhibition assays (*p* < 0.001 for all) compared to the aqueous extract. It also demonstrated lower EC50 values for DPPH, FRAP, ABTS, and COX-2 inhibition (*p* < 0.001, 0.05, 0.001, and 0.001, respectively) compared to the 95% hydroalcoholic extract. Consequently, the 50% hydroalcoholic extract was selected for formulating the 6GS used in subsequent experiments.

Table 2 compares the unencapsulated extract with the 6GS phytosome. The 6GS showed total phenolic and flavonoid contents of 320.21 ± 2.77 mg/g and 70.07 ± 2.30 mg/g of the extract, respectively. While the difference between the unencapsulated extract and 6GS was not statistically significant, the 6GS demonstrated slightly higher values. The phytosome formulation further allowed for the identification and quantification of the primary bioactive compounds, 6-gingerol and 6-shogaol. As shown in Table 2 and the HPLC profile in Figure 1, each milligram of the 50% hydroalcoholic extract contained 47.63 ± 1.98 μg/mg of 6-gingerol and 74.43 ± 1.68 μg/mg of 6-shogaol. In comparison, each milligram of the 6GS contained 51.25 ± 2.08 μg/mg of 6-gingerol and 86.33 ± 2.11 μg/mg of 6-shogaol. Notably, the concentration of 6-shogaol in the 6GS was significantly higher than in the unencapsulated extract (*p* < 0.05).

The chromatogram of the main components identified 6-gingerol and 6-shogaol as the primary compounds in both the extract and the 6GS. Their retention times were 10.851 and 21.342 min, respectively, with a peak symmetry factor of 1.0, indicating acceptable chromatographic efficiency. Furthermore, the UV spectrum exhibited significant absorbance peaks in the 200–300 nm range, consistent with the characteristic profiles of 6-gingerol and 6-shogaol. These findings confirm that 6-gingerol and 6-shogaol are the predominant bioactive compounds in both the unencapsulated extract and 6GS.

To evaluate the advantages of phytosome encapsulation, we assessed the biological activities of the 6GS. As shown in Table 2, the 6GS exhibited significantly lower EC50 values in antioxidant assays (DPPH, FRAP, and ABTS) and COX-2 suppression assays (*p* < 0.05, 0.05, 0.05, and 0.001, respectively). These results suggest that 6GS has strong bioactive potential and was thus selected for further testing in neuronal cell models.

### 2.2. Evaluation of H_2_O_2_ and 6GS Cytotoxicity on SH-SY5Y Cell Viability

H_2_O_2_-induced cytotoxicity is a well-established model known for eliciting various adverse effects, including reduced cell viability, inflammation, impaired scavenging enzyme activity, increased ROS production, and apoptosis. To establish this neurotoxic model, SH-SY5Y cells were exposed to different concentrations of H_2_O_2_ (0, 25, 50, 100, 200, 400, and 800 µM) for 24 h, followed by the assessment of cell viability using the MTT assay. Based on our previous research and relevant literature, we determined that treatment with 200 µM H_2_O_2_ for 24 h was the optimal condition for subsequent experiments [23].

The neurotoxicity of 6GS was evaluated by exposing SH-SY5Y cells to various concentrations of the extract (0, 7.8125, 15.625, 31.25, 62.5, 125, 250, 500, and 1000 µg/mL) for 24 h, as illustrated in Figure 2. The results showed a concentration-dependent decrease in cell viability: 99.64 ± 0.59, 98.60 ± 0.64, 95.43 ± 3.39, 86.61 ± 1.88, 83.29 ± 1.39, 77.91 ± 1.58, 64.08 ± 2.71, and 51.84 ± 1.91 for concentrations of 7.8125, 15.625, 31.25, 62.5, 125, 250, 500, and 1000 µg/mL, respectively. Significant reductions in cell viability were observed at concentrations of 62.5, 125, 250, 500, and 1000 µg/mL (*p* < 0.01, 0.001, 0.001, 0.001, and 0.001, respectively, compared to the control group). Consequently, doses of 15.625 and 31.25 µg/mL of 6GS were identified as the maximum non-toxic doses for SH-SY5Y cells. These doses were selected for further investigation to assess the protective effect of 6GS against H_2_O_2_-induced neurotoxicity.

### 2.3. Effect of 6GS on H_2_O_2_-Induced Cytotoxicity in SH-SY5Y Cells

The findings in Figure 3 highlight the neuroprotective properties of 6GS against H_2_O_2_-induced cytotoxicity in SH-SY5Y cells. Exposure to H_2_O_2_ alone significantly reduced cell viability (*p* < 0.01) compared to the control group, as illustrated in Figure 3b. H_2_O_2_-treated cells also showed a noticeable reduction in cell density without significant alterations in cell structure, as seen in Figure 3a. However, treatment with 6GS at concentrations of 15.625 and 31.25 µg/mL significantly reversed the decrease in cell viability (*p* < 0.01 for both concentrations) compared to cells treated with H_2_O_2_ alone. These results emphasize the potential of 6GS to protect cell viability under oxidative stress conditions.

### 2.4. Effects of 6GS on Intracellular ROS Production

Oxidative stress has critical roles in various neurodegenerative disorders. Elevated ROS levels can lead to neuronal damage. Our study aimed to examine how 6GS influences ROS production in SH-SY5Y cells subjected to H_2_O_2_-induced cytotoxicity. The results, illustrated in Figure 4, demonstrated a significant increase in ROS levels in cells treated with H_2_O_2_ and the vehicle (*p* < 0.001, compared to the control group). Notably, all doses of 6GS treatment effectively reduced ROS production (*p* < 0.001 for all, compared to the group treated with H_2_O_2_ + vehicle).

### 2.5. Effects of 6GS on the Oxidative Stress Status

The study explored the protective effects of 6GS on oxidative stress by examining MDA levels and the activity of key scavenger enzymes such as CAT, SOD, and GSH-Px as shown in Table 3. Exposure of SH-SY5Y cells to H_2_O_2_ and the vehicle resulted in a significant increase in MDA levels (*p* < 0.01 compared to the control group) and a decrease in CAT, SOD, and GSH-Px levels (*p* < 0.001 for all compared to the control group). However, treatment with 6GS at 31.25 µg/mL significantly reduced MDA levels and increased the activities of CAT, SOD, and GSH-Px (*p* < 0.05, 0.001, 0.001, and 0.001, respectively, compared to the group treated with H_2_O_2_ and the vehicle). Similarly, 6GS at 15.625 µg/mL significantly decreased MDA levels (*p* < 0.05 compared to the group treated with H_2_O_2_ and the vehicle) and increased CAT and GSH-Px levels (*p* < 0.01 and 0.001, respectively, compared to the group treated with H_2_O_2_ and the vehicle). However, there was no significant difference in SOD activity with 6GS treatment at 15.625 µg/mL.

### 2.6. Effects of 6GS on the Regulation of PI3K/Akt Pathway

Due to the crucial role of the PI3K/Akt pathway in regulating apoptosis, we examined how 6GS influences PI3K/Akt phosphorylation in SH-SY5Y cells exposed to H_2_O_2_-induced cytotoxicity, as detailed in Figure 5 and Figure 6. Our findings revealed significant insights. SH-SY5Y cells treated with H_2_O_2_ and the vehicle exhibited a notable reduction in PI3K and Akt phosphorylation (*p* < 0.001 for both compared to the control group). However, treatment with all concentrations of 6GS effectively reversed this decrease in PI3K and Akt phosphorylation induced by H_2_O_2_ (*p* < 0.01 and 0.001, respectively, for PI3K; and *p* < 0.001 for all doses, for Akt, compared to the H_2_O_2_ and vehicle group).

Importantly, no significant changes in total Akt expression were observed between the groups, confirming that the effects of 6GS on Akt activation are specific to phosphorylation rather than changes in total protein levels. These results underscore the neuroprotective role of the PI3K/Akt-signaling pathway in mediating the effects of 6GS in SH-SY5Y cells.

### 2.7. Effects of 6GS on Apoptotic Markers

The unwanted initiation of apoptosis plays a crucial role in the advancement of various neurodegenerative conditions. To investigate the neuroprotective mechanisms of 6GS, we evaluated the levels of Bcl-2, an anti-apoptotic marker, and caspase-3, an apoptotic marker. The outcomes presented in Figure 7 and Figure 8 demonstrate that SH-SY5Y cells exposed to H_2_O_2_ and the vehicle experienced a notable decline in Bcl-2 expression and a rise in caspase-3 expression (*p* < 0.001 and 0.01, respectively, compared to the control group). Significantly, treatment with all doses of 6GS effectively reinstated Bcl-2 expression (*p* < 0.001 for all doses, compared to the group treated with H_2_O_2_ + vehicle). Moreover, all doses of 6GS treatment notably suppressed caspase-3 expression (*p* < 0.001 for 15.625 µg/mL; and *p* < 0.05 for 31.25 µg/mL, compared to the group treated with H_2_O_2_ + vehicle).

### 2.8. Effects of 6GS on Inflammatory Markers

Due to the well-established role of inflammation in the progression of neurodegenerative diseases, the induction of neurotoxicity leads to the release of pro-inflammatory cytokines such as TNF-α, which directly contribute to neuronal cell death through various mechanisms, including oxidative stress and apoptosis activation. Therefore, we examined the effects of 6GS on inflammatory markers, particularly TNF-α, in SH-SY5Y cells subjected to H_2_O_2_-induced cytotoxicity, as shown in Figure 9. Our results revealed a significant increase in TNF-α expression in SH-SY5Y cells treated with H_2_O_2_ and the vehicle (*p* < 0.001, compared to the control group). Notably, this TNF-α upregulation was substantially reduced by 6GS at all administered doses (*p* < 0.001 for all doses, compared to the H_2_O_2_ + vehicle-treated group). 

## 3. Discussion

*Zingiber officinale* Roscoe has long been valued for its diverse applications in both food and traditional medicine. Among its active compounds, 6-gingerol and 6-shogaol stand out due to their notable antioxidant, anti-inflammatory, and neuroprotective effects [24]. These compounds have shown promise in addressing neurological conditions like Alzheimer’s, Parkinson’s, Huntington’s, and ALS by protecting the nervous system from oxidative stress and inflammation, two critical factors in neurodegeneration [25]. Their antioxidative properties help neutralize free radicals that contribute to cell damage, while their anti-inflammatory effects mitigate neuroinflammation by suppressing key inflammatory pathways [26,27]. Furthermore, ginger’s bioactive components may influence apoptosis, providing additional therapeutic options for neurodegeneration [26,27,28]. 

However, challenges such as poor bioavailability and stability often limit the therapeutic potential of natural bioactive compounds [29]. To address these limitations, the 6GS phytosome formulation was developed to enhance the solubility, absorption, and stability of ginger’s bioactive compounds. Our study demonstrated that although the total phenolic and flavonoid content in the unencapsulated extract and 6GS was not significantly different, the phytosome formulation achieved a significantly higher concentration of 6-shogaol (*p* < 0.05). This increase can be attributed to the greater stability and higher lipid solubility of 6-shogaol compared to 6-gingerol. Formed through the dehydration of 6-gingerol, 6-shogaol is more resistant to degradation, particularly under the extraction and encapsulation conditions used in this study. Its enhanced stability and lipid solubility allow for better integration into the lipid-rich phytosome environment, which facilitates more efficient interaction with ROS in the DPPH, FRAP, and ABTS assays. These assays, which rely on the ability of antioxidants to neutralize free radicals, demonstrated superior antioxidant activities for 6GS (all *p* < 0.05). The improved lipid solubility likely enhances the compound’s ability to scavenge ROS in lipid-rich environments, thereby boosting its antioxidant potential. Furthermore, enhanced stability ensures that 6-shogaol maintains its bioactivity throughout the assays, leading to more reliable and reproducible results.

In addition, the increased lipid solubility and stability of 6-shogaol may also play a role in its significant COX-2 suppression (*p* < 0.001). By improving the compound’s interaction with cell membranes and enzymes, the phytosome formulation potentially enhances its ability to inhibit COX-2 activity, which is involved in inflammation. These findings align with the growing body of evidence supporting the neuroprotective role of ginger’s bioactive compounds in combating oxidative stress and neuroinflammation, both hallmarks of neurodegenerative disease progression [25].

H_2_O_2_, a ROS, plays a crucial role in biological systems. At low-to-moderate concentrations, it functions as a signaling molecule involved in various physiological processes, such as cell growth, differentiation, and immune responses. However, at high concentrations, H_2_O_2_ induces oxidative stress, leading to cellular damage and even cell death, particularly in neuronal cells. When H_2_O_2_ levels exceed the capacity of the cellular antioxidant defense systems, it can cause oxidative stress by inducing the production of other ROS [30]. This excess of ROS can lead to the oxidation of lipids, proteins, and DNA, disrupting cellular function and integrity [31]. In neuronal cells, oxidative stress induced by H_2_O_2_ can trigger several pathways that ultimately lead to neurotoxicity and cell death [32].

Oxidative stress induced by H_2_O_2_ can directly oxidize cellular components such as lipids, proteins, and DNA, leading to cellular dysfunction and damage, as mentioned earlier. For instance, lipid peroxidation can disrupt cell membranes and compromise their integrity, while protein oxidation can impair enzymatic activity and disrupt cellular signaling pathways [33]. Additionally, oxidative stress induced by H_2_O_2_ can activate inflammatory pathways in neuronal cells [34]. This involves the activation of transcription factors such as NF-κB and the release of pro-inflammatory cytokines such as TNF-α [35]. These inflammatory mediators can further exacerbate oxidative stress and promote neuronal damage.

Moreover, H_2_O_2_-induced oxidative stress can trigger apoptotic cell death in neuronal cells. This involves the activation of intracellular signaling pathways such as the mitochondrial apoptotic pathway and the death receptor pathway. In the mitochondrial pathway, oxidative stress can lead to the release of cytochrome C from mitochondria, activating caspases and initiating the apoptotic cascade. In the death receptor pathway, oxidative stress can induce the activation of death receptors on the neuronal cell surface, leading to the activation of caspases and apoptosis [36,37]. Overall, H_2_O_2_-induced neurotoxicity involves a complex interplay of oxidative stress, inflammation, and apoptosis, ultimately leading to neuronal dysfunction and cell death.

Our findings substantiate that exposure to H_2_O_2_ amplifies oxidative stress through a reduction in the activities of vital scavenging enzymes such as CAT, SOD, and GSH-Px. This exposure concurrently increases the levels of MDA, a byproduct of lipid peroxidation, serving as a robust indicator of oxidative stress, along with an elevation in ROS levels. However, treatment with 6GS effectively counteracted these detrimental changes, indicating its potential to bolster the activities of scavenger enzymes while diminishing MDA and ROS levels. This reduction in oxidative stress, evidenced by the decline in MDA and ROS, significantly contributed to the amelioration of neuronal cell injury and the facilitation of neural cell survival. Remarkably, these observations align closely with prior research documented in the existing literature [23,38].

Furthermore, in our H_2_O_2_-induced neurotoxicity model, we observed the induction of apoptotic cell death through the upregulation of caspase-3 expression. However, treatment with 6GS also effectively attenuated this undesired apoptosis by downregulating the expression of caspase-3. These outcomes not only corroborate with previous investigations but also underscore the crucial role of oxidative stress and inflammation in influencing adverse apoptotic mechanisms that ultimately culminate in neuronal cell death.

The PI3K/Akt pathways intricately regulate cell survival, proliferation, and apoptosis. Activation of these pathways responds to various extracellular signals, leading to the phosphorylation and activation of Akt. Phosphorylated Akt initiates downstream events promoting cell survival and inhibiting apoptosis [39]. A crucial regulatory point in these pathways is the modulation of Bcl-2 expression and caspase-3 activity. Bcl-2, an anti-apoptotic protein primarily on the outer mitochondrial membrane, crucially regulates apoptosis. Phosphorylated Akt upregulates Bcl-2 expression, inhibiting cytochrome C release from mitochondria, a critical step in the intrinsic apoptotic pathway. Additionally, phosphorylated Akt blocks caspase-3 activity is responsible for apoptosis initiation and execution [40,41,42].

Moreover, Bcl-2 inhibits the generation of ROS, which includes pivotal signaling molecules regulating cellular processes, including apoptosis. By reducing ROS levels, Bcl-2 promotes cell survival and prevents apoptosis. Interestingly, decreased ROS levels also reduce Akt activity inhibition, forming a feedback loop reinforcing cell survival signals. In summary, PI3K/Akt pathways play a critical role in regulating cell fate decisions, particularly apoptosis, through Bcl-2 upregulation and caspase-3 activity inhibition, promoting cell survival. Modulation of ROS levels by Bcl-2 further inhibits apoptosis initiation [43,44,45].

Our data support these findings, emphasizing the potential neuroprotective effects of 6GS treatment by activating the PI3K/Akt pathway. This leads to Akt phosphorylation, upregulation of Bcl-2, and downregulation of caspase-3 expression, suppressing apoptosis and enhancing neuronal cell viability. Additionally, 6GS upregulates Bcl-2, inhibiting ROS generation and further improving neuronal cell survival.

The findings of this study hold significant promise in the context of neurodegenerative diseases, where oxidative stress and inflammation play central roles in disease progression. With conditions such as Alzheimer’s, Parkinson’s, and Huntington’s, the accumulation of oxidative damage and neuroinflammation contributes to neuronal dysfunction and degeneration. The ability of 6GS to mitigate oxidative stress, suppress neuroinflammation, and inhibit apoptotic pathways offers a potential therapeutic approach for these diseases.

Moreover, the enhancement of bioavailability and stability through the phytosome formulation addresses a critical limitation in the clinical application of natural compounds. This formulation could offer more effective delivery and sustained bioactivity of ginger’s bioactive components, making it a promising candidate for developing adjunctive therapies aimed at reducing the burden of neurodegenerative diseases. The modulatory effect on the PI3K/Akt pathway, which is pivotal in cell survival and apoptosis regulation, further highlights the therapeutic potential of 6GS. By activating neuroprotective mechanisms such as Bcl-2 expression and reducing ROS levels, 6GS may offer a novel approach to protecting neurons from the damaging effects of oxidative stress and inflammation in a range of neurodegenerative disorders.

In conclusion, this study provides compelling evidence that 6GS, particularly in its phytosome formulation, holds promise as a therapeutic agent for neurodegenerative diseases. By addressing critical pathophysiological factors such as oxidative stress, neuroinflammation, and apoptosis, 6GS offers a multifaceted approach to neuroprotection. Future clinical studies are needed to further explore its potential as part of a broader therapeutic strategy aimed at mitigating the progression of neurodegenerative diseases.

## 4. Materials and Methods

### 4.1. Preparation of 6GS

The 6-gingerol- and 6-shogaol-enriched extract was derived from the rhizomes of *Zingiber officinale* Roscoe specimens sourced from Loei Province, Northeastern Thailand, an area known for its stringent cultivation standards, overseen by the Department of Agricultural Extension of Thailand. The plant specimen was authenticated by a pharmacognosy expert at the National Museum of Thai Traditional Medicine, where it is cataloged under voucher specimen No. 0002402. Following authentication, the plant material was thoroughly cleaned and dried at 60 °C for 48 h in a Memmert GmbH oven (Eagle, WI, USA). The dried material was then finely pulverized into a powder.

For extraction, the powdered material was macerated sequentially with distilled water, a 50% hydro-alcoholic solution, and a 95% hydro-alcoholic solution. The resulting extract was centrifuged at 3000× *g* for 10 min and filtered using Whatman No. 1 filter paper. Ethanol was subsequently removed from the filtrate using rotary evaporation, and the residue was dried with a Labconco freeze dryer (Kansas City, MO, USA). The sequential maceration method was chosen to optimize the extraction of bioactive compounds with varying polarities, including the primary bioactives 6-gingerol and 6-shogaol.

Phytosome encapsulation was performed based on methods outlined in previous studies [46]. Briefly, the *Zingiber officinale* extract was dissolved in 50% ethanol and selected for phytosome preparation due to its effectiveness in extracting both polar and non-polar bioactive compounds, including 6-gingerol and 6-shogaol. This solvent demonstrated the highest potential in preserving active ingredients and biological activities among various extracts. Phosphatidylcholine was dissolved in dichloromethane, and the two solutions were mixed and sonicated at a frequency of 25–30 kHz for 2 min, repeated three times. The mixture was stirred at 25 °C for 8 h, followed by high-speed centrifugation to facilitate the formation of lipid vesicles. Ethanol and dichloromethane were removed using a rotary evaporator at 45 °C for 3 h. The final solution was dried using a BUCHI Mini Spray Dryer (B-290, BÜCHI Labortechnik AG, Flawil, Switzerland) and stored in a desiccator containing silica gel at 4 °C.

The resulting 6GS phytosome was utilized to quantify bioactive compounds, assess antioxidant and anti-inflammatory activities, and apply them in neuronal cell treatment experiments.

### 4.2. Analysis of the Chromatographic Fingerprints

To identify and quantify the unencapsulated extract and 6GS, we established a chromatographic fingerprint using high-performance liquid chromatography (HPLC) with a Waters^®^ system equipped with a Waters^®^ 2998 photodiode array detector (Milford, MA, USA). Separation was accomplished using Purospher^®^ STAR, C-18 encapped (5 μM), LiChro-CART^®^ 250-4.6, and HPLC-Cartridge, Sorbet Lot No. HX255346 (Merck, Darmstadt, Germany).

For the mobile phase gradient, we utilized 100% methanol (solvent A) (Fisher Scientific, Waltham, MA, USA) and 2.5% acetic acid (solvent B) (Fisher Scientific, USA) in deionized (DI) water. The gradient elution was performed at a flow rate of 1.0 mL/min according to the following profile: 0–16 min, 70% A; 17–19 min, 100% A; 20–25 min, 10% A. Before injection, the sample underwent filtration (0.45 μm, Millipore, Burlington, MA, USA), and a 20 μL aliquot of the unencapsulated extract and 6GS were directly used. The chromatogram was analyzed at 280 nm using a UV detector, and EmpowerTM3 was employed for data analysis [23,46].

### 4.3. Analysis of Total Phenolic Compounds and Flavonoids

We assessed the total phenolic compound content in the sample using the Folin–Ciocalteu colorimetric method and a microplate reader (iMark™ Microplate Absorbance Reader, Bio-Rad Laboratories, Hercules, CA, USA). Initially, 20 µL of sample were mixed with a freshly prepared solution containing 20 µL of 50% *v*/*v* Folin–Ciocalteu reagent (Sigma–Aldrich, St. Louis, MO, USA) and 158 µL of distilled water. Following an 8 min incubation period, 30 µL of 20% Na_2_CO_3_ (Sigma–Aldrich, USA) were introduced. Subsequently, the mixture underwent further incubation in a dark environment at 25 °C for 2 h, after which its absorbance was measured at 765 nm (BioTek Synergy H1 Multimode Reader, BioTek Synergy H1 Multimode Reader, Winooski, VT, USA). A standard calibration curve was prepared using gallic acid (Sigma–Aldrich, USA) [23,46].

To determine the total flavonoid content in the sample, we employed the aluminum chloride method. Specifically, 100 µL of sample at various concentrations were combined with 100 µL of 2% methanolic aluminum chloride (Sigma–Aldrich, USA). The resulting mixture was then left to incubate at 25 °C in a dark environment for 30 min, after which its absorbance at 415 nm was measured against a suitable blank. A standard calibration curve was prepared using quercetin (Sigma–Aldrich, USA) [23,46].

### 4.4. Analysis of Biological Activities

#### 4.4.1. DPPH

The free-radical-scavenging activity of the samples was evaluated using the 1,1-diphenyl-2-picrylhydrazyl (DPPH) assay. A methanolic solution of DPPH was prepared at a concentration of 0.1 mM and mixed with 0.3 mL of the samples at varying concentrations in a 2 mL reaction volume. After a 30 min incubation at 25 °C, absorbance was recorded at 517 nm using a BioTek Synergy H1 Multimode Reader (USA) [23,46].

#### 4.4.2. FRAP

The ferric-reducing antioxidant power (FRAP) was determined using the reduction of ferric 2,4,6-tripyridyl-s-triazine (Fe^3+^-TPTZ) to ferrous 2,4,6-tripyridyl-s-triazine (Fe^2+^-TPTZ). A freshly prepared FRAP reagent, consisting of 20 mM of ferric chloride (FeCl_3_), 300 mM of acetate buffer, and 10 mM of 2,4,6-tripyridyl-s-triazine (TPTZ), was mixed in a 1:10:1 ratio. Then, 10 µL of the sample were combined with 90 µL of the reagent. Following a 10 min incubation at 37 °C, absorbance was measured at 593 nm against a blank using the BioTek Synergy H1 Multimode Reader (USA) [23,46,47].

#### 4.4.3. ABTS

The free-radical-scavenging activity was assessed using 2,2-azinobis-(3-ethylbenzothiazoline-6-sulfonic acid) (ABTS). An ABTS radical cation (ABTS^•+^) solution was generated by mixing 7 mM of ABTS with 2.45 mM of potassium persulfate (K_2_S_2_O_8_) in a 2:3 ratio and allowing the mixture to react overnight. For the assay, 30 µL of the sample were diluted with 120 µL of distilled water and 30 µL of ethanol, followed by the addition of 3 mL of ABTS^•+^ solution. Absorbance was measured at 734 nm using a Pharmacia LKB-Biochrom 4060 spectrophotometer (Uppsala, Sweden) [23,46,48].

#### 4.4.4. COX-2

The inhibition of Cyclo-oxygenase-2 (COX-2) activity was evaluated using a colorimetric COX-2 inhibitor screening assay kit, following the manufacturer’s instructions. The COX-2 working solution was prepared by dissolving COX-2 in 100 mM of Tris(hydroxymethyl)aminomethane hydrochloride (Tris-HCl) buffer, pH 8.0, at a 1:100 ratio. The reaction mixture, comprising assay buffer, sample, heme, COX-2 working solution, N,N,N′,N′-tetramethyl-p-phenylenediamine (TMPD), and arachidonic acid, was added to a 96-well microplate. After incubation at 25 °C for 30 min, absorbance was recorded at 590 nm using a BioTek Synergy H1 Multimode Reader (USA). Indomethacin was used as the reference standard.

In all instances, results were expressed as EC50, representing the half maximal effective concentration.

### 4.5. Cell Culture and Cell Viability Assay

#### 4.5.1. Cell Culture

The SH-SY5Y cell line, derived from a neuroblastoma bone marrow biopsy and exhibiting neuronal characteristics, was obtained from the American Type Culture Collection (ATCC, catalog number CRL-2266, Manassas, VA, USA). The cells were cultured in Dulbecco’s Modified Eagle Medium (DMEM, Gibco, Waltham, MA, USA) supplemented with 10% fetal bovine serum (FBS), 1% penicillin-streptomycin, and 1% non-essential amino acids. Cultures were maintained at 37 °C in a humidified atmosphere containing 5% carbon dioxide (CO_2_). For each experiment, cells were plated at a density appropriate for the assay being conducted. Before treatment, the culture medium was aspirated, and a fresh medium containing H_2_O_2_, with or without 6GS, was added [23,38].

#### 4.5.2. Cell Viability Assessment

To evaluate the in vitro cytotoxicity of H_2_O_2_ and 6GS, the MTT assay was employed. SH-SY5Y cells were seeded in 96-well plates at a density of 1 × 10^4^ cells/well and cultured under the aforementioned conditions. Cells were exposed to varying concentrations of H_2_O_2_ (ranging from 0 µM to 800 µM) and 6GS (ranging from 0 µg/mL to 1000 µg/mL) in serum-free DMEM for 24 h. The concentration of H_2_O_2_-induced cytotoxicity was optimized to match our previous studies [23,38], where the optimal concentration for inducing toxicity was determined to be 200 µM for 24 h based on cell viability measurements. 

To investigate the potential protective effects of 6GS against H_2_O_2_-induced neurotoxicity, cells were pre-treated with the 6GS for 24 h. Subsequently, fresh medium containing H_2_O_2_, with or without 6GS, was added. After 24 h of treatment, the culture medium was replaced with MTT reagent from Sigma, USA, at a final concentration of 0.5 mg/mL. Cells were incubated at 37° for 1 h in a humidified atmosphere with a 5% CO_2_ atmosphere. Following incubation, the MTT reagent was aspirated, and 100 µL of dimethyl sulfoxide (DMSO) were added to dissolve the insoluble purple formazan product. The resulting solution’s absorbance was measured at 570 nm using a microplate reader (BioTek Synergy H1 Multimode Reader, USA) [23,38].

### 4.6. Analysis of Intracellular Reactive Oxygen Species Levels

Intracellular ROS levels were assessed using 5-(and-6)-chloromethyl-2′,7′-dichlorodihydrofluorescein diacetate acetyl ester (CM-H_2_DCFDA), a cell-permeable fluorescent probe. SH-SY5Y cells were seeded into a 96-well plate and cultured as described previously. After 24 h of incubation, cells were pretreated with or without 6GS. The cells were then incubated with 10 μM of CM-H_2_DCFDA at 37 °C in a 5% CO_2_ incubator under dark conditions for 30 min. Following this, cells were washed with phosphate-buffered saline (PBS) and exposed to H_2_O_2_ in serum-free medium for 24 h. Intracellular esterases converted CM-H_2_DCFDA into 2′,7′-dichlorofluorescein (DCF) in the presence of ROS, resulting in fluorescence. The fluorescence intensity, reflective of ROS levels, was measured using a Synergy HT Multi-Mode Microplate Reader (BioTek Instruments, Winooski, VT, USA) with excitation at 488 nm and emission at 520 nm [23,38]. 

### 4.7. Analysis of Oxidative Stress Changes

Cellular homogenates were prepared to assess changes in oxidative stress markers. Cells were homogenized in 0.1 M of potassium phosphate (KH_2_PO_4_) buffer (pH 7.4), with 1 mg of the sample diluted in 20 µL of PBS. The protein concentration in the homogenates was quantified using a Thermo Scientific NanoDrop 2000c spectrophotometer (Wilmington, DE, USA) by measuring optical absorbance at 280 nm [46]. 

#### 4.7.1. Catalase

Catalase activity was measured by assessing its ability to decompose H_2_O_2_. A reaction mixture containing 10 µL of the prepared sample, 25 µL of 4 M sulfuric acid (H_2_SO_4_), 50 µL of 30 mM H_2_O_2_ in 50 mM phosphate buffer (pH 7.0), and 150 µL of 5 mM potassium permanganate (KMnO_4_) were prepared. The absorbance was recorded at 490 nm. Catalase standards (10–100 units/mL) were used for comparison [49].

#### 4.7.2. Superoxide Dismutase

Superoxide dismutase activity was measured using a method adapted from Sun et al. A reaction mixture was prepared comprising 0.5 mM of xanthine (pH 7.4), 0.2 M of KH_2_PO_4_ buffer (pH 7.8), 0.01 M of ethylenediaminetetraacetic acid (EDTA), and 15 µM of cytochrome C in a 50:25:1:1 (*v*/*v*) ratio. The prepared sample (20 µL) was mixed with 200 µL of the reaction mixture and 20 µL of 0.90 mU/mL xanthine oxidase. The absorbance was measured at 415 nm, with superoxide dismutase standards (0–25 units/mL) used for reference [50].

#### 4.7.3. Glutathione Peroxidase

The activity of glutathione peroxidase was evaluated by combining 20 µL of the prepared sample with a reaction mixture containing 10 µL of 1 mM dithiothreitol (DTT), 10 mM of monosodium phosphate (NaH_2_PO_4_) in distilled water, 1 mM of sodium azide (NaN_3_) in 40 mM of potassium phosphate buffer (K_3_PO_4_) at pH 7.0, 10 µL of 50 mM glutathione (GSH) solution, and 100 µL of 30% H_2_O_2_. After incubation at 25 °C for 10 min, 10 µL of 10 mM of 5,5′-dithiobis-(2-nitrobenzoic acid) (DTNB) was added, and absorbance at 412 nm was measured. Glutathione peroxidase enzyme standards (1–5 units/mL) were used for reference [51].

#### 4.7.4. Malondialdehyde

The malondialdehyde level, an indicator of lipid peroxidation, was quantified using the thiobarbituric acid (TBA) reaction. A reaction mixture containing 10 µL of the prepared sample, 10 µL of 8.1% sodium dodecyl sulfate (SDS), 75 µL of 0.8% TBA, 75 µL of 20% acetic acid (CH_3_COOH), and 30 µL of distilled water were prepared. The mixture was heated at 95 °C for 10 min, then cooled to room temperature. After cooling, 250 µL of a mixture of *n*-butanol and pyridine (15:1 ratio) and 50 µL of distilled water were added. The solution was centrifuged at 4000× *g* for 10 min, and the upper layer was collected. The absorbance was measured at 532 nm. Standards of 1,1,3,3-tetramethoxypropane (TMP), ranging from 0 to 15 µM, were used for calibration [52].

### 4.8. Western Blot Analysis

Proteins were extracted from SH-SY5Y cells using an RIPA buffer (Cell Signaling Technology, Danvers, MA, USA) prepared with Tris-HCl (20 mM, pH 7.5), NaCl (150 mM), Na_2_EDTA (1 mM), EGTA (1 mM), NP-40 (1%), sodium deoxycholate (1%), sodium pyrophosphate (2.5 mM), β-glycerophosphate (1 mM), Na_3_VO_4_ (1 mM), leupeptin (1 µg/mL), and PMSF (1 mM). The mixture was centrifuged at 10,000× *g* at 4 °C for 10 min, and the supernatant was collected. Protein concentrations were determined using a NanoDrop 2000c spectrophotometer (Thermo Fisher Scientific, Waltham, MA, USA).

For separation, 20 µg of protein lysate were combined with loading buffer, denatured at 95 °C for 5 min, and subjected to SDS-PAGE. Proteins were then transferred onto a PVDF membrane, blocked with 5% skimmed milk in TBS-T (0.1%), and incubated overnight at 4 °C with primary antibodies specific to PI3K, p-Akt, Akt, Bcl-2, caspase-3, TNF-α, or β-actin (Cell Signaling Technology, USA; dilution range: 1:1000–1:2000).

After washing with TBS-T, the membrane was incubated with HRP-conjugated anti-rabbit IgG (1:2000 dilution) for 1 h at room temperature. Protein bands were detected using the Clarity™ Western ECL Substrate (Bio-Rad, Hercules, CA, USA, cat. no. 170-5060) and visualized with a ChemiDoc™ MP system, utilizing Image Lab software (version 6.0.0; Bio-Rad).

Cropped Western blot images focused on target proteins are included in the manuscript, while full-length blots are provided in Appendix A. Data were expressed as relative densities compared to the control group [23,46,48].

### 4.9. Statistical Analysis

The results are presented as the mean ± standard error of the mean (SEM). Statistical significance was determined using a one-way analysis of variance (ANOVA), followed by the post hoc Tukey test for multiple comparisons. For comparisons between two groups, the student’s *t*-test was applied. A significance level of *p* < 0.05 was considered statistically significant. All statistical analyses were conducted using SPSS version 28 (IBM Corp., Armonk, NY, USA, Released 2022, IBM SPSS Statistics for Mac OS).

## 5. Conclusions

This study highlights the neuroprotective effects of 6GS against H_2_O_2_-induced neurotoxicity in SH-SY5Y cells. The findings demonstrate that 6GS mitigates oxidative stress by enhancing the activity of antioxidant enzymes, suppressing neuroinflammation, and inhibiting apoptosis through the PI3K/Akt-signaling pathway, as shown in Figure 10. These mechanisms collectively promote neuronal survival and highlight the potential of 6GS as functional ingredients for developing neuroprotective interventions targeting neurodegenerative diseases.

Due to the promising in vitro results, further research is essential to evaluate the safety profile of 6GS through comprehensive toxicity studies. Such efforts will pave the way for subsequent in vivo investigations and clinical trials, advancing the potential application of 6GS in dietary supplements or therapeutic formulations.

## Figures and Tables

**Figure 1 molecules-29-06046-f001:**
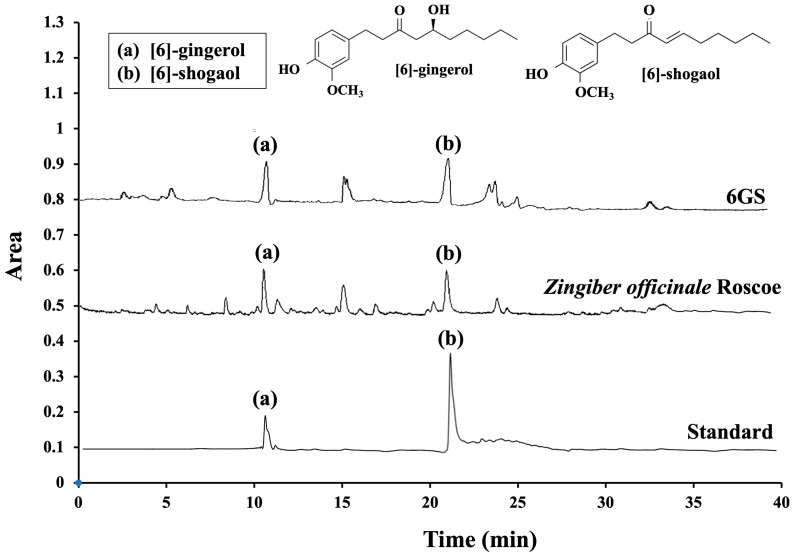
Chromatographic fingerprinting of primary derivatives of *Zingiber officinale* Roscoe extract and 6GS; 6GS: phytosome-encapsulated 6-gingerol- and 6-shogaol-enriched extracts from *Zingiber officinale* Roscoe.

**Figure 2 molecules-29-06046-f002:**
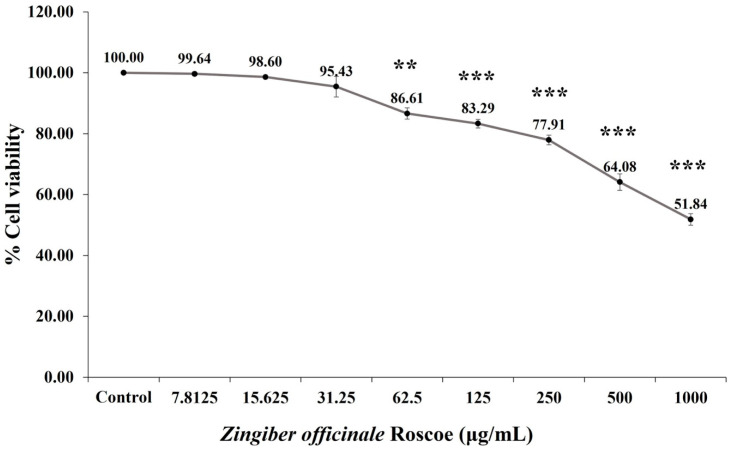
The effect of 6GS on the viability of SH-SY5Y cells. Data are presented as mean ± SEM. **^,^*** *p* < 0.01 and 0.001, respectively, compared to the control group; 6GS: phytosome-encapsulated 6-gingerol- and 6-shogaol-enriched extracts from *Zingiber officinale* Roscoe.

**Figure 3 molecules-29-06046-f003:**
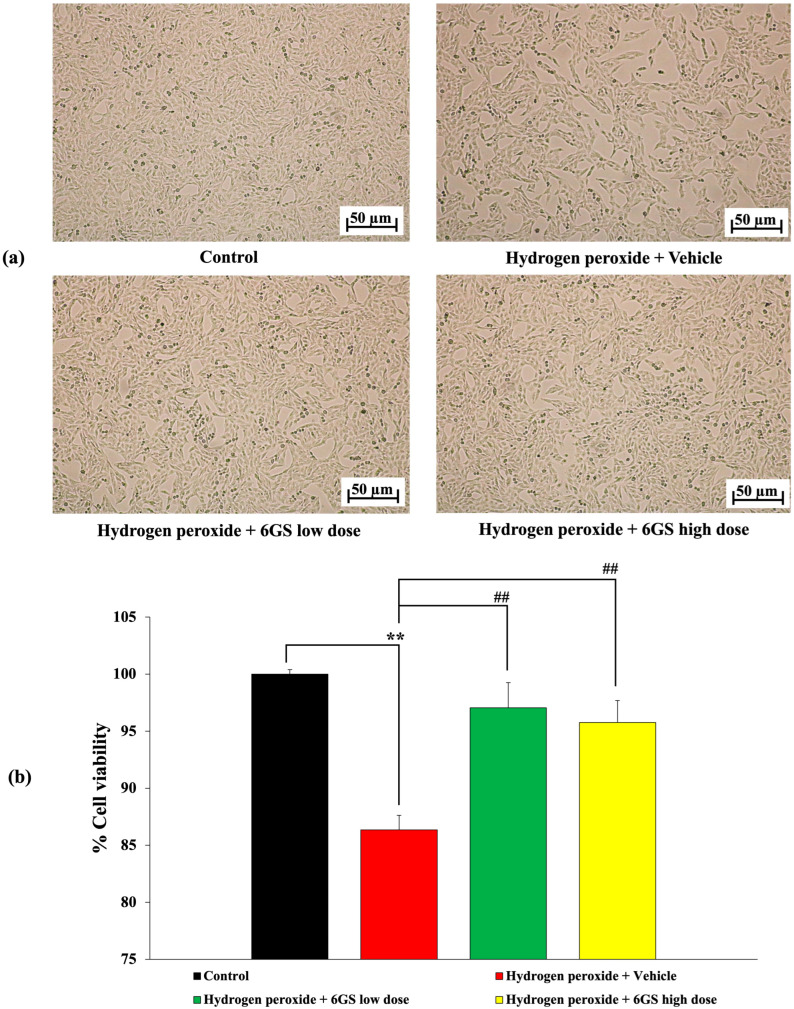
Effect of 6GS on H_2_O_2_-induced cytotoxicity in SH-SY5Y cells. (**a**) Light microscopy of SH-SY5Y cell density at 10× magnification. (**b**) Percentage of cell viability of SH-SY5Y cells. ** *p* < 0.01, compared to the control; ^##^ *p* < 0.01, compared to SH-SY5Y cells treated with H_2_O_2_ and vehicle. H_2_O_2_: hydrogen peroxide at a dose of 200 μM; 6GS low dose and 6GS high dose: phytosome-encapsulated 6-gingerol- and 6-shogaol-enriched extracts from *Zingiber officinale* Roscoe at doses of 15.625 and 31.25 μg/mL, respectively.

**Figure 4 molecules-29-06046-f004:**
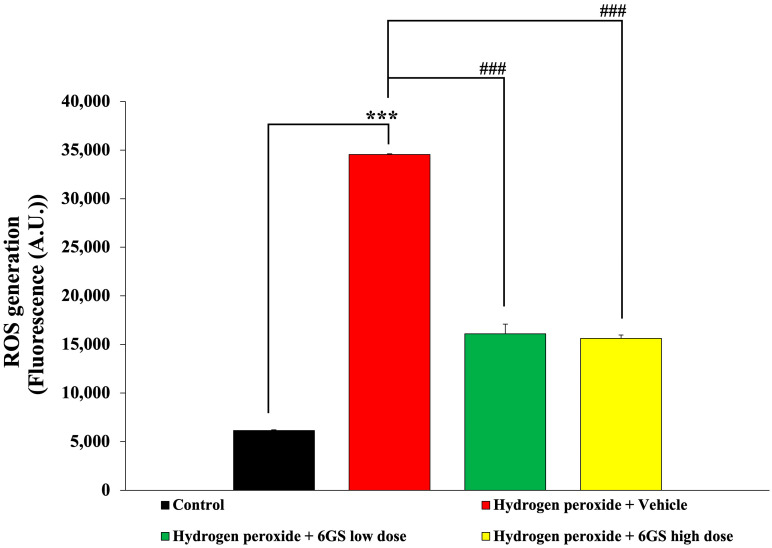
Effect of 6GS on ROS generation in SH-SY5Y cell toxicity induced by H_2_O_2_. Data are presented as mean ± SEM. *** *p* < 0.001, compared to the control; ^###^ *p* < 0.001, compared to SH-SY5Y cells treated with H_2_O_2_ and vehicle. H_2_O_2_: hydrogen peroxide at a dose of 200 μM; 6GS low dose and 6GS high dose: phytosome-encapsulated 6-gingerol- and 6-shogaol-enriched extracts from *Zingiber officinale* Roscoe at doses of 15.625 and 31.25 μg/mL, respectively.

**Figure 5 molecules-29-06046-f005:**
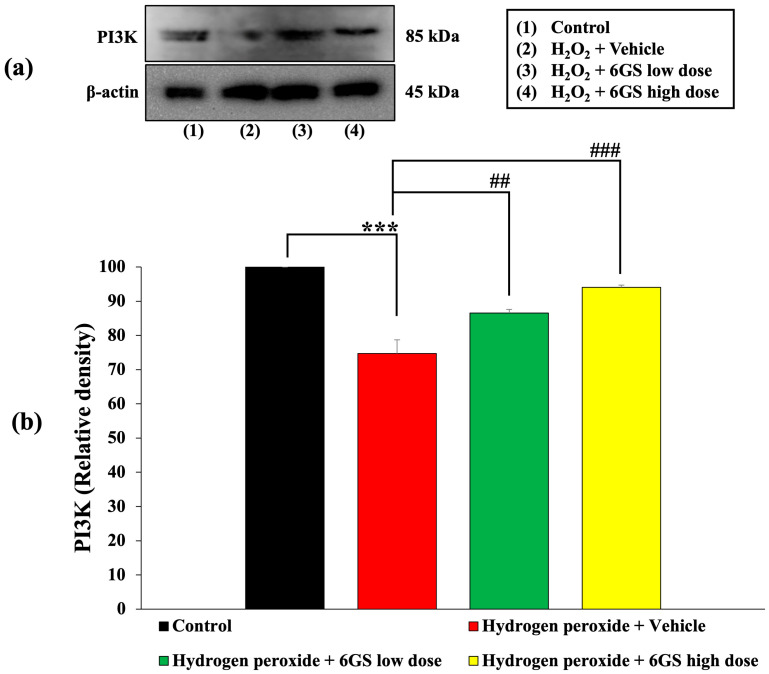
Effect of 6GS on the expression of PI3K in H_2_O_2_-induced SH-SY5Y cell toxicity was detected by Western blotting. (**a**) Representative Western blot showing the levels of PI3K. (**b**) Relative density of PI3K. Data are presented as mean ± SEM. *** *p* < 0.001, compared to the control; ^##,###^ *p* < 0.01 and 0.001, respectively, compared to SH-SY5Y cells treated with H_2_O_2_ and vehicle. H_2_O_2_: hydrogen peroxide at a dose of 200 μM; 6GS low dose and 6GS high dose: phytosome-encapsulated 6-gingerol- and 6-shogaol-enriched extracts from *Zingiber officinale* Roscoe at doses of 15.625 and 31.25 μg/mL, respectively.

**Figure 6 molecules-29-06046-f006:**
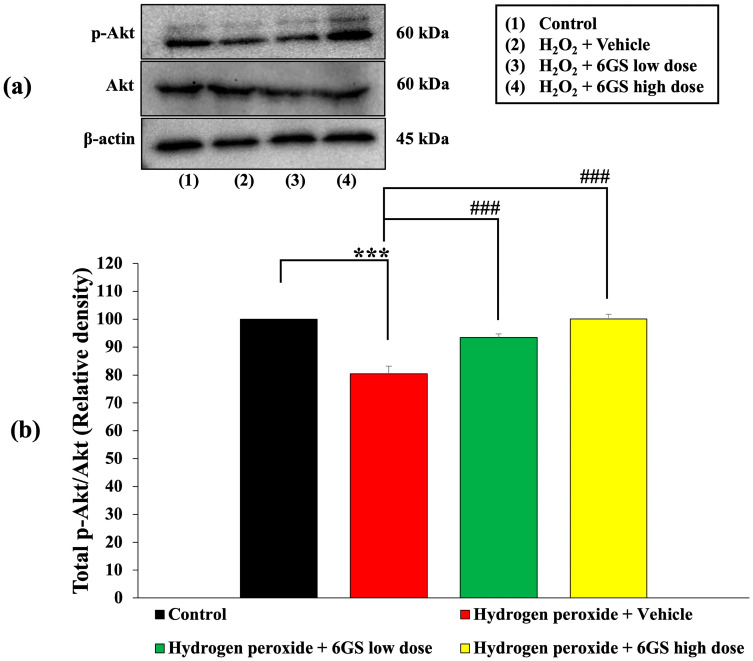
Effect of 6GS on the expression of p-Akt/Akt in H_2_O_2_-induced SH-SY5Y cell toxicity was detected by Western blotting. (**a**) Representative Western blot showing the levels of p-Akt/Akt. (**b**) Relative density of p-Akt/Akt. Data are presented as mean ± SEM. *** *p* < 0.001, compared to the control; ^###^
*p* < 0.001, compared to SH-SY5Y cells treated with H_2_O_2_ and vehicle. H_2_O_2_: hydrogen peroxide at a dose of 200 μM; 6GS low dose and 6GS high dose: phytosome-encapsulated 6-gingerol- and 6-shogaol-enriched extracts from *Zingiber officinale* Roscoe at doses of 15.625 and 31.25 μg/mL, respectively.

**Figure 7 molecules-29-06046-f007:**
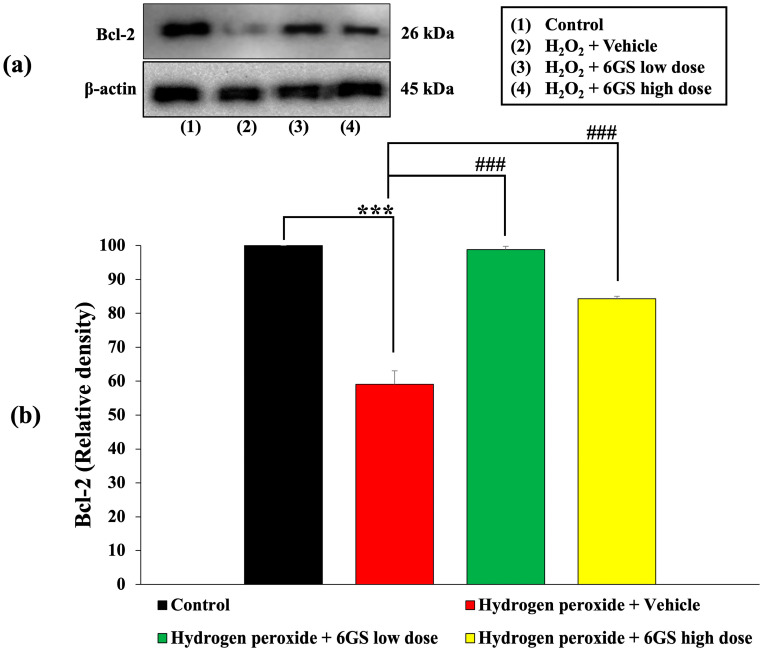
Effect of 6GS on the expression of Bcl-2 in H_2_O_2_-induced SH-SY5Y cell toxicity was detected by Western blotting. (**a**) Representative Western blot showing the levels of Bcl-2. (**b**) Relative density of total Bcl-2. Data are presented as mean ± SEM. *** *p* < 0.001, compared to the control; ^###^
*p* < 0.001, compared to SH-SY5Y cells treated with H_2_O_2_ and vehicle. H_2_O_2_: hydrogen peroxide at a dose of 200 μM; 6GS low dose and 6GS high dose: phytosome-encapsulated 6-gingerol- and 6-shogaol-enriched extracts from *Zingiber officinale* Roscoe at doses of 15.625 and 31.25 μg/mL, respectively.

**Figure 8 molecules-29-06046-f008:**
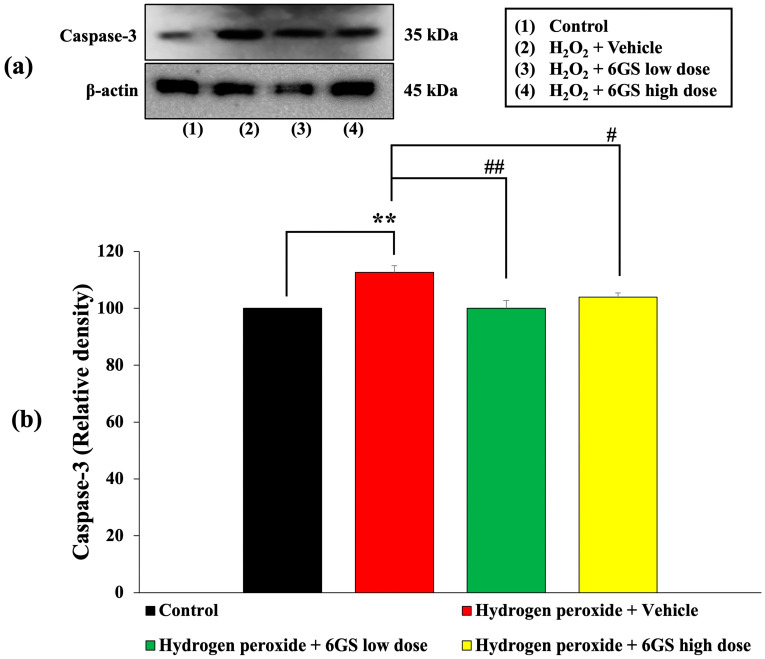
Effect of 6GS on the expression of caspase-3 in H_2_O_2_-induced SH-SY5Y cell toxicity was detected by Western blotting. (**a**) Representative Western blot showing the levels of caspase-3. (**b**) Relative density of caspase-3. Data are presented as mean ± SEM. ** *p* < 0.01, compared to the control; ^#,##^
*p* < 0.05 and 0.01, respectively, compared to SH-SY5Y cells treated with H_2_O_2_ and vehicle. H_2_O_2_: hydrogen peroxide at a dose of 200 μM; 6GS low dose and 6GS high dose: phytosome-encapsulated 6-gingerol- and 6-shogaol-enriched extracts from *Zingiber officinale* Roscoe at doses of 15.625 and 31.25 μg/mL, respectively.

**Figure 9 molecules-29-06046-f009:**
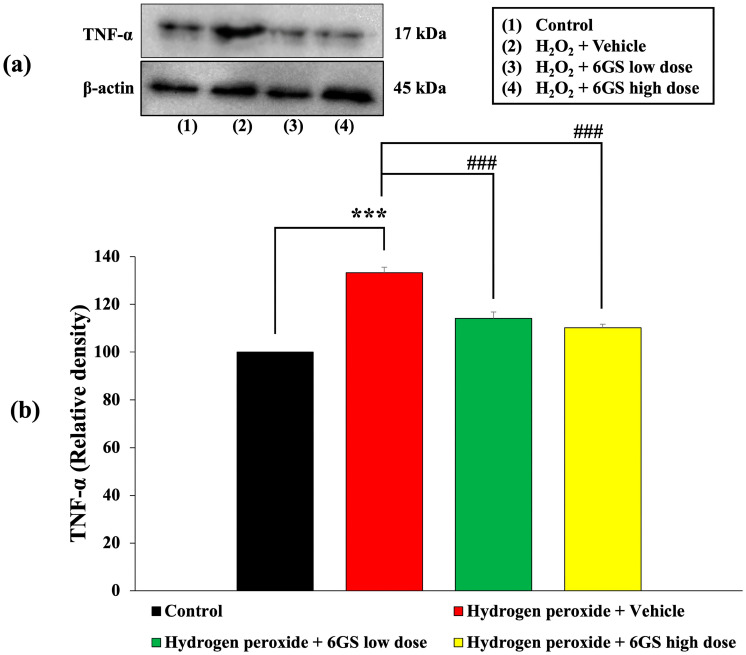
Effect of 6GS on the expression of TNF-α in H_2_O_2_-induced SH-SY5Y cell toxicity was detected by Western blotting. (**a**) Representative Western blot showing the levels of TNF-α. (**b**) Relative density of TNF-α. Data are presented as mean ± SEM. *** *p* < 0.001, compared to the control; ^###^
*p* < 0.001, compared to SH-SY5Y cells treated with H_2_O_2_ and vehicle. H_2_O_2_: hydrogen peroxide at a dose of 200 μM; 6GS low dose and 6GS high dose: phytosome-encapsulated 6-gingerol- and 6-shogaol-enriched extracts from *Zingiber officinale* Roscoe at doses of 15.625 and 31.25 μg/mL, respectively.

**Figure 10 molecules-29-06046-f010:**
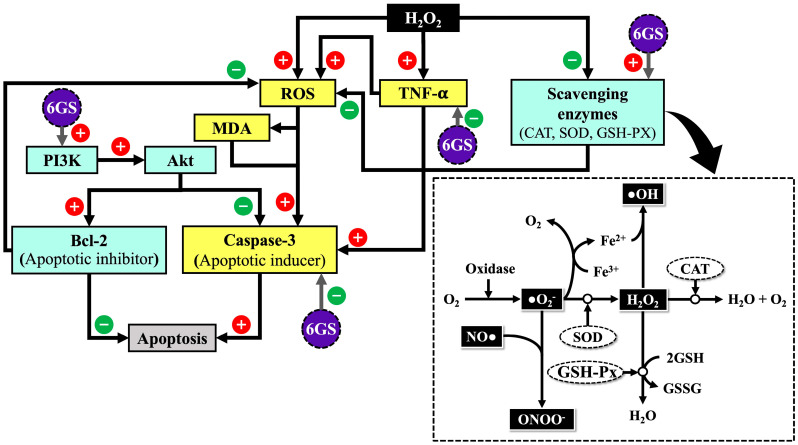
The schematic diagram demonstrates the neuroprotective potential of 6GS in H_2_O_2_-induced SH-SY5Y neurotoxicity; 6GS: phytosome-encapsulated 6-gingerol- and 6-shogaol-enriched extracts from *Zingiber officinale* Roscoe; H_2_O_2_: hydrogen peroxide; •O^2−^: superoxide anion; •OH: hydroxyl radical; NO•: nitric oxide; ONOO^−^: peroxynitrit; ROS: reactive oxygen species; SOD: superoxide dismutase; CAT: catalase; GSH-Px: glutathione peroxidase; MDA: malondialdehyde; PI3K: phosphoinositide 3-kinase; Akt: protein kinase B; Bcl-2: B-cell lymphoma 2; and TNF-α: tumor necrosis factor-alpha.

**Table 1 molecules-29-06046-t001:** Bioactive compounds and biological activities among the various extracts of *Zingiber officinale* Roscoe.

Parameters	Units	Aqueous Extract	50% Hydro-Alcoholic Extract	95% Hydro-Alcoholic Extract
Bioactive compounds				
Total phenolic content	mg Gallic acid/g extract	207.60 ± 2.25	315.43 ± 3.01 ***^,###^	251.48 ± 0.42 ***
Total flavonoid content	mg Quercetin/g extract	25.22 ± 0.48	68.11 ± 0.48 ***^,###^	47.72 ± 0.08 ***
Biological activities				
DPPH	EC50 μg/mL	70.04 ± 5.20	17.69 ± 3.45 ***^,###^	60.86 ± 4.02
FRAP	EC50 μg/mL	64.89 ± 5.22	26.87 ± 4.05 ***^,#^	40.98 ± 1.72 ***
ABTS	EC50 μg/mL	102.48 ± 2.91	24.82 ± 2.26 ***^,###^	69.35 ± 1.86 ***
COX-2	EC50 μg/mL	48.36 ± 0.09	28.43 ± 0.42 ***^,###^	47.40 ± 0.24

Data are presented as mean ± SEM. *** *p* < 0.001, compared with the aqueous extract; ^#,###^
*p* < 0.05 and 0.001, respectively, compared with the 95% hydro-alcoholic extract.

**Table 2 molecules-29-06046-t002:** Comparison of bioactive compounds and biological activities of *Zingiber officinale* Roscoe extract and 6GS.

Parameters	Units	*Zingiber officinale* Roscoe Extract	6GS
Bioactive compounds			
Total phenolic content	mg Gallic acid/g extract	315.43 ± 3.01	320.21 ± 2.77
Total flavonoid content	mg Quercetin/g extract	68.11 ± 0.48	70.03 ± 2.30
Ginger derivatives			
6-gingerol	μg/mg extract	47.63 ± 1.98	51.25 ± 2.08
6-shogaol	μg/mg extract	74.43 ± 1.68	83.33 ± 1.01 *
Biological activities			
DPPH	EC50 μg/mL	17.69 ± 3.45	12.02 ± 0.05 *
FRAP	EC50 μg/mL	26.87 ± 4.05	19.24 ± 1.02 *
ABTS	EC50 μg/mL	24.82 ± 2.26	17.34 ± 0.53 *
COX-2	EC50 μg/mL	28.43 ± 0.42	14.23 ± 0.29 ***

Data are presented as mean ± SEM. *^,^*** *p* < 0.05 and 0.001, respectively, when comparing *Zingiber officinale* Roscoe extracts with 6GS; 6GS: phytosome-encapsulated 6-gingerol and 6-shogaol-enriched extracts from *Zingiber officinale* Roscoe.

**Table 3 molecules-29-06046-t003:** The effect of the 6GS on oxidative stress markers in SH-SY5Y cell cytotoxicity induced by H_2_O_2_.

Treatment Groups	MDA (ng/mg Protein)	CAT (Units/mg Protein)	SOD (Units/mg Protein)	GSH-Px (Units/mg Protein)
Control	16.14 ± 2.85	27.59 ± 1.61	24.23 ± 0.55	25.25 ± 0.36
H_2_O_2_ + Vehicle	32.82 ± 3.41 **	11.90 ± 1.11 ***	11.44 ± 0.40 ***	11.97 ± 0.58 ***
H_2_O_2_ + 6GS low dose	19.45 ± 0.22 ^#^	23.92 ± 1.63 ^##^	11.73 ± 0.55	20.69 ± 0.84 ^###^
H_2_O_2_ + 6GS high dose	19.24 ± 0.41 ^#^	27.28 ± 0.63 ^###^	18.29 ± 0.97 ^###^	21.37 ± 0.99 ^###^

Data are presented as mean ± SEM. **^,^*** *p* < 0.01 and 0.001, respectively, compared to control; ^#,##,###^ *p* < 0.05, 0.01, and 0.001, respectively, compared to SH-SY5Y cells, which received hydrogen peroxide and vehicle; 6GS low dose and 6GS high dose: phytosome-encapsulated 6-gingerol- and 6-shogaol-enriched extracts from *Zingiber officinale* Roscoe at doses of 15.625 and 31.25 μg/mL, respectively.

## Data Availability

The data that support the findings of this study are available from the corresponding author upon reasonable request.

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
