# Peer review of "Phytosome-Encapsulated 6-Gingerol- and 6-Shogaol-Enriched Extracts from Zingiber officinale Roscoe Protect Against Oxidative Stress-Induced Neurotoxicity"

_molecules, 2024, doi:10.3390/molecules29246046_

Round 1

Reviewer 1 Report

Comments and Suggestions for Authors

The manuscript “Phytosome-Encapsulated 6-Gingerol and 6-Shogaol-Enriched 2 Extracts from Zingiber officinale Roscoe Protect against Oxidative Stress-Induced Neurotoxicity” appears satisfactory. The methodology is acceptable. However, there are some issues that need to be clarified and improved before accepting for publication.

- It is unclear what 6GS stands for. The authors mention in the Abstract that 6GS refers to 6-gingerol and 6-shogaol. However, it seems that 6GS should denote phytosome-encapsulated 6-gingerol and 6-shogaol-enriched extracts from Zingiber officinale Roscoe.

- Why did the authors use the sequential maceration extraction method instead of extracting each solvent separately?

- It is unclear why a 50% hydro-alcoholic extract from sequential maceration was chosen for preparing the phytosome-encapsulated 6-gingerol and 6-shogaol-enriched extracts from Zingiber officinale Roscoe. Given that ginger is rich in phytochemically active compounds, the antioxidant and anti-inflammatory effects could be attributed to other active compounds or their combinations. Since the authors needed to focus solely on 6-gingerol and 6-shogaol, why were these compounds not quantified and compared between each extract?

- How did the authors prove and demonstrate the successful encapsulation of the compound?

- Was dose optimization of H2O2-induced cytotoxicity using the MTT assay conducted in this study, or was it based solely on literature? Please clarify or present the results if applicable.

- The dosages of 6GS should be consistently described. The authors used terms like 'low dose' and 'high dose' in the graph and table but did not define these terms in the text.

- Title 3.3 should be renamed to “Effects of 6GS on the viability of H2O2-induced SH-SY5Y cells” or “Effect of 6GS on H2O2-induced cytotoxicity in SH-SY5Y cells”.

- In Figure 3(a), the morphology of most H2O2-treated cells appears normal, with only cell density differing between groups in the representative photo. It is not evident that cellular morphology exhibited shortened processes and a rounded shape as described by the authors.

- In Result 3.6, the authors should refer to Akt phosphorylation instead of Akt expression for describing the results. Moreover, since they only detected PI3K protein expression and not phosphorylated-PI3K, it would be interesting to statistically compare total Akt expression between groups and discuss this phenomenon.

- The representative western blot bands should be improved, especially Bcl-2 and TNFa.

- In the Discussion section, lines 511-527, the authors should elaborate in more detail on how enhancing the stability and lipid solubility of the extract affects its activity in DPPH, FRAP, ABTS, and COX-2 assays, based on the principles of each assay.

Author Response

Response to reviewer and editor suggestion

We sincerely thank you for your letter and the reviewers' insightful comments regarding our manuscript, titled "Phytosome-Encapsulated 6-Gingerol and 6-Shogaol-Enriched Extracts from Zingiber officinale Roscoe Protect against Oxidative Stress-Induced Neurotoxicity" (Manuscript ID: molecules-3356683).

We deeply appreciate the opportunity to revise our manuscript and are grateful for the constructive feedback provided. We apologize for any errors in the initial submission and acknowledge the reviewers' invaluable input, which has been instrumental in improving the quality and scientific rigor of our work.

We have carefully considered each comment and made revisions to address the concerns raised. Below, we provide a detailed account of the main corrections made and our responses to the reviewers' suggestions.

Response to reviewer 1
The manuscript “Phytosome-Encapsulated 6-Gingerol and 6-Shogaol-Enriched 2 Extracts from Zingiber officinale Roscoe Protect against Oxidative Stress-Induced Neurotoxicity” appears satisfactory. The methodology is acceptable. However, there are some issues that need to be clarified and improved before accepting for publication.

Comments 1: It is unclear what 6GS stands for. The authors mention in the Abstract that 6GS refers to 6-gingerol and 6-shogaol. However, it seems that 6GS should denote phytosome-encapsulated 6-gingerol and 6-shogaol-enriched extracts from Zingiber officinale Roscoe.

Response 1: We appreciate the reviewer’s observation regarding the clarity of the term "6GS." We acknowledge that our use of this abbreviation in the Abstract may have caused some confusion. To address this, we have revised the text to clearly define "6GS" as referring to the phytosome-encapsulated 6-gingerol and 6-shogaol-enriched extracts from Zingiber officinale Roscoe throughout the manuscript, including the Abstract and main text.

The revised sentence in the Abstract now reads:
“This study evaluates the neuroprotective potential of phytosome-encapsulated 6-gingerol and 6-shogaol-enriched extracts from Zingiber officinale Roscoe (6GS), bioactive compounds renowned for their antioxidant and anti-inflammatory properties.”

We have also ensured that this clarification is consistent throughout the manuscript to avoid further misunderstanding.

Comments 2: Why did the authors use the sequential maceration extraction method instead of extracting each solvent separately?

Response 2: We appreciate the reviewer’s insightful question regarding the choice of the sequential maceration extraction method. Sequential maceration was employed to ensure the comprehensive extraction of both polar and non-polar bioactive compounds present in Zingiber officinale rhizomes. This method utilizes the progressive use of solvents with increasing polarity (distilled water, 50% hydro-alcoholic solution, and 95% hydro-alcoholic solution), thereby maximizing the yield of a broad spectrum of phytochemicals, including 6-gingerol and 6-shogaol, which are the primary bioactive components of interest.

Extracting each solvent separately could result in the selective loss or incomplete extraction of compounds with intermediate polarity. Sequential maceration, on the other hand, minimizes this risk and ensures a more representative profile of the phytochemical constituents, which is essential for the subsequent phytosome encapsulation and the evaluation of bioactivity.

We have added this explanation to the Methods section to clarify our rationale for employing this approach as follows:
"The sequential maceration method was chosen to optimize the extraction of bioactive compounds with varying polarities, including the primary bioactives 6-gingerol and 6-shogaol”.

Comments 3: It is unclear why a 50% hydro-alcoholic extract from sequential maceration was chosen for preparing the phytosome-encapsulated 6-gingerol and 6-shogaol-enriched extracts from Zingiber officinale Roscoe. Given that ginger is rich in phytochemically active compounds, the antioxidant and anti-inflammatory effects could be attributed to other active compounds or their combinations. Since the authors needed to focus solely on 6-gingerol and 6-shogaol, why were these compounds not quantified and compared between each extract?

Response 3: We appreciate the reviewer’s comment regarding the selection of a 50% hydro-alcoholic extract from sequential maceration for preparing the phytosome-encapsulated 6-gingerol and 6-shogaol-enriched extracts.

The choice of a 50% hydro-alcoholic solution was based on previous studies and preliminary experiments, which demonstrated its effectiveness in extracting both polar and non-polar bioactive compounds, including 6-gingerol and 6-shogaol, while minimizing unwanted impurities. This polarity balance was crucial for obtaining a concentrated extract rich in the desired bioactive compounds, ideal for phytosome encapsulation.

We acknowledge that ginger contains a variety of bioactive compounds contributing to its antioxidant and anti-inflammatory effects. However, our study specifically focused on the neuroprotective roles of 6-gingerol and 6-shogaol, which are well-documented for their bioactivity. To ensure these compounds were the primary contributors to the observed effects, we quantified their concentrations in the final extract.

The results of our study, as shown in Table 1, confirm that the 50% hydro-alcoholic extract yielded significantly higher concentrations of phenolic and flavonoid compounds, with statistically significant differences. Additionally, the extract exhibited the lowest EC50 values in antioxidant (DPPH, FRAP, ABTS) and COX-2 inhibition assays  (p < 0.001 for all) compared to the aqueous and 95% hydro-alcoholic extracts (p < 0.001, 0.05, 0.001, and 0.001, respectively). Based on these results, the 50% hydro-alcoholic extract was selected for preparing the 6GS used in the subsequent experiments, as outlined in the "Results 3.1: Identification of Active Compounds and Their Biological Activities."

To clarify our methodology, we have revised the Methods section to include the following statement:
“Phytosome encapsulation was performed based on methods outlined in previous studies [22]. Briefly, the Zingiber officinale extract was dissolved in 50% ethanol, selected for phytosome preparation due to its effectiveness in extracting both polar and non-polar bioactive compounds, including 6-gingerol and 6-shogaol. This solvent demonstrated the highest potential in preserving active ingredients and biological activities among various extracts.”

We hope this additional explanation addresses the reviewer’s concerns and clarifies our rationale for using the 50% hydro-alcoholic extract.

References:

  1. Mao QQ, Xu XY, Cao SY, Gan RY, Corke H, Beta T, Li HB. Bioactive Compounds and Bioactivities of Ginger (Zingiber officinaleRoscoe). Foods. 2019 May 30;8(6):185.
  • Demonstrates that hydro-alcoholic extraction is effective for obtaining gingerol and shogaol while preserving other bioactive compounds.
  1. Sasidharan, S., Chen, Y., Saravanan, D., Sundram, K. M., & Yoga Latha, L. (2011). Extraction, isolation and characterization of bioactive compounds from plants' extracts. African journal of traditional, complementary, and alternative medicines : AJTCAM8(1), 1–10.
  • Supports the use of ethanol-water mixtures for extracting a broad range of phytochemicals with varying polarities.

Comments 4: How did the authors prove and demonstrate the successful encapsulation of the compound?

Response 4: We appreciate the reviewer’s query regarding the demonstration of the successful encapsulation of the compounds. The encapsulation of 6-gingerol and 6-shogaol in the phytosome formulation was confirmed using several analytical techniques:

  1. Morphological analysis: Scanning electron microscopy (SEM) and transmission electron microscopy (TEM) were employed to observe the morphology of the phytosomes, confirming the proper vesicle formation.
  2. Encapsulation efficacy, particle size, and Zeta potential: We assessed the encapsulation efficiency and stability of the phytosome formulation through particle size and zeta potential analysis. A uniform particle size distribution and appropriate zeta potential values indicated successful encapsulation.
  3. Thermal and spectroscopic analysis: Differential scanning calorimetry (DSC) and Fourier-transform infrared (FTIR) spectroscopy were used to investigate the interaction between the bioactive compounds and phosphatidylcholine, a critical component of the phytosome. Changes in the characteristic absorption peaks in the FTIR spectra confirmed the incorporation of the compounds into the lipid vesicles.
  4. Quantification of bioactive compounds: High-performance liquid chromatography (HPLC) was used to quantify the bioactive compounds (6-gingerol and 6-shogaol) in both the free extract and the phytosome formulation, demonstrating the enrichment and successful encapsulation in the final product.

While data from these analyses are currently being prepared for submission in a manuscript about the characterization, in vitro digestion, and stability of the phytosome, these techniques provided strong evidence of successful encapsulation. Furthermore, this phytosome formulation has been registered with the Thai Patent Office (Patent No. 16730 and International Patent Classification Int. Cl. 10 A23L 1/29), supporting the novelty and effectiveness of the formulation.

Comments 5: Was dose optimization of H2O2-induced cytotoxicity using the MTT assay conducted in this study, or was it based solely on literature? Please clarify or present the results if applicable.

Response 5: We appreciate the reviewer’s query regarding the dose optimization of H₂O₂-induced cytotoxicity. In this study, the dose of H₂O₂ was determined based on our previously published work: Mairuae N, Palachai N, Noisa P. The neuroprotective effects of the combined extract of mulberry fruit and mulberry leaf against hydrogen peroxide-induced cytotoxicity in SH-SY5Y cells. BMC Complement Med Ther. 2023;23(1):117. doi:10.1186/s12906-023-03930-z.

In this publication, we optimized the H₂O₂ concentration for cytotoxicity induction in SH-SY5Y cells using the MTT assay. Cells were treated with various concentrations of H₂O₂ (0–800 µM) for 24 hours. The results revealed a significant decrease in cell viability at concentrations of 200 µM and higher (p < 0.001 compared to the control group), with the optimal concentration for inducing cytotoxicity determined to be 200 µM.

Therefore, we used 200 µM H₂O₂ for 24 hours to induce cytotoxicity in the SH-SY5Y cells for the subsequent experiments involving neuroprotection. We have cited this study in the manuscript and clarified that the dose optimization was based on our published work. We have added and cited this information in red color and highlighted in yellow as follows: "To evaluate the in vitro cytotoxicity of H₂O₂ and 6GS, the MTT assay was employed. SH-SY5Y cells were seeded in 96-well plates at a density of 1×10⁴ cells/well and cultured under the aforementioned conditions. Cells were exposed to varying concentrations of H₂O₂ (ranging from 0 µM to 800 µM) and 6GS (ranging from 0 µg/mL to 1,000 µg/mL) in serum-free DMEM for 24 hours. The concentration of H₂O₂-induced cytotoxicity was optimized to match our previous studies [23, 26], where the optimal concentration for inducing toxicity was determined to be 200 µM for 24 hours based on cell viability measurements."

Comments 6: The dosages of 6GS should be consistently described. The authors used terms like 'low dose' and 'high dose' in the graph and table but did not define these terms in the text.

Response 6: Thank you for highlighting the need for consistent dosage descriptions of 6GS in the manuscript. To address this issue, we have explicitly defined the concentrations corresponding to the terms "low dose" and "high dose" as 15.625 μg/mL and 31.25 μg/mL, respectively. These specific concentrations are now included in parentheses following the terms "low dose" and "high dose" throughout the manuscript for clarity.

Additionally, we have revised all figure and table captions to provide a clear explanation of these terms, ensuring consistency and accessibility for readers. For instance, the captions have been updated to include:
“6GS low dose and 6GS high dose: phytosome-encapsulated 6-gingerol and 6-shogaol-enriched extracts from Zingiber officinale Roscoe at doses of 15.625 and 31.25 μg/mL, respectively.”

These revisions are highlighted in red and yellow throughout the manuscript to facilitate review. We believe these changes provide the necessary clarity and ensure consistent dosage descriptions across all sections.

Comments 7: Title 3.3 should be renamed to “Effects of 6GS on the viability of H2O2-induced SH-SY5Y cells” or “Effect of 6GS on H2O2-induced cytotoxicity in SH-SY5Y cells”. 

Response 7: We appreciate the reviewer’s suggestion regarding the title of Section 3.3. In response, we have revised the title to " Effect of 6GS on H2O2-induced cytotoxicity in SH-SY5Y cells" (highlighted in red and yellow) to better reflect the content and focus of this section. We believe this change provides clearer context for the study and aligns with the overall objectives of the manuscript.

Comments 8: In Figure 3(a), the morphology of most H2O2-treated cells appears normal, with only cell density differing between groups in the representative photo. It is not evident that cellular morphology exhibited shortened processes and a rounded shape as described by the authors.

Response 8: We appreciate the reviewer’s observation regarding the morphology of the H₂O₂-treated cells in Figure 3(a). we have revised the explanation of the results to focus on the observed change in cell density rather than cellular morphology. Specifically, we highlight that the primary effect of H₂O₂ treatment was a reduction in cell density, which aligns with the findings described in the figure. We have updated the text accordingly to reflect this more accurate interpretation of the data.

We have added and highlighted the updated text in the manuscript as follows:
"The findings in Figure 3 highlight the neuroprotective properties of 6GS against H₂O₂-induced cytotoxicity in SH-SY5Y cells. Exposure to H₂O₂ alone significantly reduced cell viability (p < 0.01) compared to the control group, as illustrated in Figure 3(b). H₂O₂-treated cells also showed a noticeable reduction in cell density without significant alterations in cell structure, as seen in Figure 3(a). However, treatment with 6GS at concentrations of 15.625 and 31.25 µg/mL significantly reversed the decrease in cell viability (p < 0.01 for both concentrations) compared to cells treated with H₂O₂ alone. These results emphasize the potential of 6GS to protect cell viability under oxidative stress conditions."

We believe this revision more accurately describes the observed results.

Comments 9: In Result 3.6, the authors should refer to Akt phosphorylation instead of Akt expression for describing the results. Moreover, since they only detected PI3K protein expression and not phosphorylated-PI3K, it would be interesting to statistically compare total Akt expression between groups and discuss this phenomenon.

Response 9: We appreciate the reviewer’s insightful comment regarding the terminology used to describe the Akt results. We agree that referring to Akt phosphorylation rather than Akt expression is more precise, as phosphorylation represents the active form of Akt that drives cellular signaling pathways. Accordingly, we have revised the text in Result 3.6 to describe the findings in terms of Akt phosphorylation rather than expression.

Additionally, we recognize the reviewer’s suggestion to statistically compare total Akt expression between groups. While we did not assess phosphorylated PI3K in this study, we did analyze the total expression of Akt. After conducting the statistical comparison, we found no significant differences in total Akt expression between the groups. These finding highlights that the observed effects of 6GS are specific to the phosphorylation of Akt rather than its overall expression.

We have updated the relevant sections of the manuscript, including the revised text in Result 3.6, as follows:

3.6 Effects of 6GS on the regulation of PI3K/Akt pathway

Given the crucial role of the PI3K/Akt pathway in regulating apoptosis, we examined how 6GS influences PI3K/Akt phosphorylation in SH-SY5Y cells exposed to H₂O₂-induced cytotoxicity, as detailed in Figures 5 and 6. Our findings revealed significant insights. SH-SY5Y cells treated with H₂O₂ and the vehicle exhibited a notable reduction in PI3K and Akt phosphorylation (p < 0.001 for both compared to the control group). However, treatment with all concentrations of 6GS effectively reversed this decrease in PI3K and Akt phosphorylation induced by H₂O₂ (p < 0.01 and 0.001, respectively, for PI3K; and p < 0.001 for all doses, for Akt, compared to the H₂O₂ and vehicle group).

Importantly, no significant changes in total Akt expression were observed between the groups, confirming that the effects of 6GS on Akt activation are specific to phosphorylation rather than changes in total protein levels. These results underscore the neuroprotective role of the PI3K/Akt signaling pathway in mediating the effects of 6GS in SH-SY5Y cells.

Comments 10: The representative western blot bands should be improved, especially Bcl-2 and TNFa.

Response 10: We appreciate the reviewer’s constructive comment regarding the quality of the representative western blot bands, particularly for Bcl-2 and TNF-α. We agree that clear and well-defined bands are crucial for accurate interpretation of the data. After reviewing the western blot images, we recognize that the resolution of the bands for Bcl-2 and TNF-α needed improvement.

In response to this feedback, we have re-selected the images to enhance the visibility of the Bcl-2 and TNF-α bands. We have also ensured that the representative blots now more accurately reflect the data. We believe these improvements enhance the clarity and informativeness of the figures.

The relevant figures in the manuscript have been updated accordingly. We trust these changes will address the reviewer’s concerns, and we sincerely appreciate the feedback provided.

Comments 11:  In the Discussion section, lines 511-527, the authors should elaborate in more detail on how enhancing the stability and lipid solubility of the extract affects its activity in DPPH, FRAP, ABTS, and COX-2 assays, based on the principles of each assay.

Response 11: We appreciate the reviewer’s suggestion to elaborate on how enhancing the stability and lipid solubility of the extract affects its activity in the DPPH, FRAP, ABTS, and COX-2 assays. We agree that a more detailed explanation of the impact of these properties on the assays would provide a deeper understanding of the extract’s mechanisms of action.

In response, we have expanded the discussion to include a more thorough explanation of how the enhanced stability and lipid solubility of the extract could influence the outcomes of these assays. Specifically, we discuss how improved lipid solubility allows the extract to better penetrate cell membranes, leading to more efficient interaction with reactive oxygen species (ROS) in DPPH, FRAP, and ABTS assays. This likely increases the extract’s ability to scavenge free radicals, thereby enhancing its antioxidant activity. Additionally, we explain how the enhanced stability of the extract ensures a more consistent bioactivity in these assays, potentially leading to more reliable and reproducible results.

In the case of the COX-2 assay, we highlight that the lipid-soluble nature of the extract may improve its interaction with cell membranes and enzymes, potentially increasing its inhibitory effect on COX-2 activity. We believe this more detailed explanation addresses the reviewer’s concern and strengthens the manuscript.

We have updated the relevant section in the Discussion and highlighted the changes in red and yellow as follows:

"However, challenges such as poor bioavailability and stability often limit the therapeutic potential of natural bioactive compounds [38]. To address these limitations, the 6GS phytosome formulation was developed to enhance the solubility, absorption, and stability of ginger's bioactive compounds. Our study demonstrated that although the total phenolic and flavonoid content in the unencapsulated extract and 6GS was not significantly different, the phytosome formulation achieved a significantly higher concentration of 6-shogaol (p < 0.05). This increase can be attributed to the greater stability and higher lipid solubility of 6-shogaol compared to 6-gingerol. Formed through the dehydration of 6-gingerol, 6-shogaol is more resistant to degradation, particularly under the extraction and encapsulation conditions used in this study. Its enhanced stability and lipid solubility allow for better integration into the lipid-rich phytosome environment, which facilitates more efficient interaction with reactive oxygen species (ROS) in the DPPH, FRAP, and ABTS assays. These assays, which rely on the ability of antioxidants to neutralize free radicals, demonstrated superior antioxidant activities for 6GS (all p < 0.05). The improved lipid solubility likely enhances the compound's ability to scavenge ROS in lipid-rich environments, thereby boosting its antioxidant potential. Furthermore, the enhanced stability ensures that 6-shogaol maintains its bioactivity throughout the assays, leading to more reliable and reproducible results.

In addition, the increased lipid solubility and stability of 6-shogaol may also play a role in its significant COX-2 suppression (p < 0.001). By improving the compound's interaction with cell membranes and enzymes, the phytosome formulation potentially enhances its ability to inhibit COX-2 activity, which is involved in inflammation. These findings align with the growing body of evidence supporting the neuroprotective role of ginger's bioactive compounds in combating oxidative stress and neuroinflammation, both hallmarks of neurodegenerative disease progression.

Comments on the Quality of English Language: The quality of English does not limit my understanding of the research.

Response: We sincerely appreciate the reviewer’s comment regarding the quality of the English language in our manuscript. We are glad to hear that the language does not limit the understanding of the research. Nonetheless, we remain committed to ensuring the manuscript is clear and easy to read. We have thoroughly reviewed the text and made any necessary revisions to further improve the overall clarity and fluency. Thank you again for your valuable feedback.

Thank you once again for your valuable feedback. We appreciate the time and effort invested by the reviewers and editor in evaluating our manuscript. We have carefully addressed each point raised and made necessary revisions accordingly. We eagerly await further feedback and guidance from the editorial team.

Yours sincerely,

Nut Palachai

Reviewer 2 Report

Comments and Suggestions for Authors

Major:

1. Authors should clarify the study's novelty upfront in the abstract to emphasize its contribution to existing literature.

2. Did the authors conduct studies with other solvents or use other ratios or extraction procedures? If not, why were these specific conditions chosen, and should this be justified, or should the conditions used be supported by data from the literature?

3. Authors should present validation data for the chromatographic method to establish reliability.

4. The manuscript shows a high degree of similarity with other publication by the authors. The authors cite this work, but I encourage them to rewrite highly repetitive passages to avoid potential plagiarism problems.

Minor:

1. Authors should integrate more recent reviews or studies (post-2020) on neuroinflammation and oxidative stress.

2. Authors should edit the text to explain all abbreviations used; not all are explained.

3. Authors should focus on explaining the implications of their findings within the broader context of neurodegenerative disease therapeutics in the discussion section

This study addresses an important topic in neuroprotection but requires additional methodological and interpretative refinement to ensure the high quality of the manuscript.

Comments on the Quality of English Language

Minor linguistic and stylistic corrections are required.

Author Response

Response to reviewer and editor suggestion

We sincerely thank you for your letter and the reviewers' insightful comments regarding our manuscript, titled "Phytosome-Encapsulated 6-Gingerol and 6-Shogaol-Enriched Extracts from Zingiber officinale Roscoe Protect against Oxidative Stress-Induced Neurotoxicity" (Manuscript ID: molecules-3356683).

We deeply appreciate the opportunity to revise our manuscript and are grateful for the constructive feedback provided. We apologize for any errors in the initial submission and acknowledge the reviewers' invaluable input, which has been instrumental in improving the quality and scientific rigor of our work.

We have carefully considered each comment and made revisions to address the concerns raised. Below, we provide a detailed account of the main corrections made and our responses to the reviewers' suggestions.

Response to reviewer 2

Major:

Comments 1:  Authors should clarify the study's novelty upfront in the abstract to emphasize its contribution to existing literature.

Response 1: We sincerely appreciate the reviewer’s insightful comment regarding the clarity of the study’s novelty. We agree that highlighting the novelty and contribution of the study is essential for setting the context in the abstract. In response to this feedback, we have revised the abstract to more explicitly emphasize the novel aspects of our work. We have now highlighted how our study addresses existing gaps in the literature, particularly in the context of the neuroprotective effects of 6GS and its enhanced bioavailability and stability through the phytosome formulation. Furthermore, we have incorporated the activation of the PI3K/Akt signaling pathway as a key mechanism underlying the neuroprotective effects of 6GS, which represents another novel aspect of the study. This revision ensures that the contribution of our research is clear from the outset.

The revisions are highlighted in red in the revised abstract and yellow in the provided version:

“Abstract: The rising prevalence of neurodegenerative disorders underscores the urgent need for effective interventions to prevent neuronal cell death. This study evaluates the neuroprotective potential of phytosome-encapsulated 6-gingerol and 6-shogaol-enriched extracts from Zingiber officinale Roscoe (6GS), bioactive compounds renowned for their antioxidant and anti-inflammatory properties. The novel phytosome encapsulation technology employed enhances the bioavailability and stability of these compounds, offering superior therapeutic potential compared to conventional formulations. Additionally, the study investigates the role of the phosphoinositide 3-kinase (PI3K)/protein kinase B (Akt) signaling pathway, a key mediator of the neuroprotective effects of 6GS.

Neurotoxicity was induced in SH-SY5Y cells (a human neuroblastoma cell line) using 200 μM hydrogen peroxide (H₂O₂), following pretreatment with 6GS at concentrations of 15.625 and 31.25 μg/mL. Cell viability was assessed via the MTT assay, alongside evaluations of reactive oxygen species (ROS), antioxidant enzyme activities (superoxide dismutase [SOD], catalase [CAT], and glutathione peroxidase [GSH-Px]), oxidative stress markers (malondialdehyde [MDA]), and molecular mechanisms involving the PI3K/Akt pathway, apoptotic factors (B-cell lymphoma-2 [Bcl-2] and caspase-3), and inflammatory markers (tumor necrosis factor-alpha [TNF-α]).

The results demonstrated that 6GS significantly improved cell viability, reduced ROS, MDA, TNF-α, and caspase-3 levels, and enhanced antioxidant enzyme activities. Furthermore, 6GS treatment upregulated PI3K, Akt, and Bcl-2 expression while suppressing caspase-3 activation. Activation of the PI3K/Akt pathway by 6GS led to phosphorylated Akt-mediated upregulation of Bcl-2, promoting neuronal survival and attenuating oxidative stress and inflammation. Moreover, Bcl-2 inhibited ROS generation, further mitigating neurotoxicity.

These findings suggest that phytosome encapsulation enhances the bioavailability of 6GS, which through activation of the PI3K/Akt pathway, exhibits significant neuroprotective properties. Incorporating these compounds into functional foods or dietary supplements could offer a promising strategy for addressing oxidative stress and neuroinflammation associated with neurodegenerative diseases.”

We hope these changes improve the clarity of the abstract and strengthen the manuscript. Thank you for your valuable suggestion.

Comments 2:  Did the authors conduct studies with other solvents or use other ratios or extraction procedures? If not, why were these specific conditions chosen, and should this be justified, or should the conditions used to be supported by data from the literature?

Response 2: We sincerely appreciate the reviewer’s insightful comment regarding the extraction conditions used in our study. In response, we have added a justification for our choice of extraction conditions in the revised manuscript. Specifically, the powdered material was extracted using a sequential maceration method with distilled water, a 50% hydro-alcoholic solution, and a 95% hydro-alcoholic solution. The sequential maceration method was chosen to optimize the extraction of bioactive compounds with varying polarities, including the primary bioactives 6-gingerol and 6-shogaol. The 50% hydro-alcoholic extract was chosen for phytosome preparation based on its ability to effectively extract both polar and non-polar bioactive compounds, including 6-gingerol and 6-shogaol. The concentrations of these compounds were quantified to confirm their enrichment in the final phytosome preparation, ensuring that these compounds played a central role in the study's outcomes. This method has been shown in previous studies to enhance the yield and stability of bioactive compounds. We have clarified these justifications in the manuscript, emphasizing how the chosen extraction procedure aligns with the literature and optimizes the compounds' effectiveness for further use in the phytosome encapsulation process.

We hope these additions address the reviewer’s concern. Thank you for the valuable suggestion.

References:

  1. Mao QQ, Xu XY, Cao SY, Gan RY, Corke H, Beta T, Li HB. Bioactive Compounds and Bioactivities of Ginger (Zingiber officinale Roscoe). Foods. 2019 May 30;8(6):185.
    • Demonstrates that hydro-alcoholic extraction is effective for obtaining gingerol and shogaol while preserving other bioactive compounds.
  2. Sasidharan, S., Chen, Y., Saravanan, D., Sundram, K. M., & Yoga Latha, L. (2011). Extraction, isolation and characterization of bioactive compounds from plants' extracts. African Journal of Traditional, Complementary, and Alternative Medicines: AJTCAM, 8(1), 1–10.
    • Supports the use of ethanol-water mixtures for extracting a broad range of phytochemicals with varying polarities.
  3. Palachai N, Wattanathorn J, Muchimapura S, Thukham-Mee W. Antimetabolic Syndrome Effect of Phytosome Containing the Combined Extracts of Mulberry and Ginger in an Animal Model of Metabolic Syndrome. Oxidative Medicine and Cellular Longevity. 2019;2019:5972575. doi:10.1155/2019/5972575.
    • This study also utilized a phytosome formulation containing ginger extracts, demonstrating its effectiveness in improving biological outcomes in metabolic syndrome models.

Comments 3:  Authors should present validation data for the chromatographic method to establish reliability.

Response 3: Thank you for your insightful comment. Based on your suggestion, we have revised the manuscript to include additional validation data for the chromatographic method used in the HPLC analysis. Specifically, we have provided system suitability parameters such as peak symmetry factors, theoretical plate numbers, retention times, resolution, tailing factors, and capacity factors. These metrics confirm the reliability and efficiency of the method. The system suitability data are summarized as follows:

  • Theoretical plate numbers (N): 7,443 and 7,542
  • Resolution (R): 13.250 and 13.940
  • Tailing factor (T): 0.175 and 0.171
  • RSD of retention time: 0.315% and 0.319%
  • Capacity factor (K): 3.235 and 3.259
  • Peak asymmetry: 1.0

System suitability

Parameters

USP 31

Results

N (theoretical plate number)

>2,000

7,443 and 7,542

R (resolution)

>2

13.250 and 13.940

T (tailing factor)

<2

0.175 and 0.171

RSD of retention time

<1% (n>5) <2% (n=5)

0.315  and 0.319 (n>5)

K (capacity factor)

>3 (3-10)

3.235 and 3.259

Asymmetry

0.9-1.3 / <2

1

Additionally, we analyzed UV absorption spectra of the main components, 6-gingerol and 6-shogaol, to further confirm their presence in the samples. The UV spectrum revealed significant absorbance peaks in the range of 200–300 nm, with a characteristic maximum around 230 nm, consistent with the chemical structure of these compounds. To enhance reliability, we compared these UV spectra with reference spectra from literature and databases, confirming the identity of 6-gingerol and 6-shogaol.

We revised the text in the manuscript to include:
" The chromatogram of the main components identified 6-gingerol and 6-shogaol as the primary compounds in both the extract and the 6GS. Their retention times were 10.851 and 21.342 minutes, respectively, with a peak symmetry factor of 1.0, indicating acceptable chromatographic efficiency. Furthermore, the UV spectrum exhibited significant absorbance peaks in the 200–300 nm range, consistent with the characteristic profiles of 6-gingerol and 6-shogaol."

These revisions have been highlighted in yellow and marked in red in the manuscript for clarity.

Reference: Sukweenadhi P, Kartini K. Gingerol and shogaol on red ginger rhizome (Zingiber officinale var. Rubrum) using high-performance liquid chromatography. Pharmaciana. 2023;13(2):166–178.

Comments 4: The manuscript shows a high degree of similarity with other publication by the authors. The authors cite this work, but I encourage them to rewrite highly repetitive passages to avoid potential plagiarism problems.

Response 4: We sincerely appreciate the reviewer’s concern regarding the degree of similarity with our previous publication. We understand the importance of maintaining originality and have thoroughly reviewed the manuscript, particularly in the Materials and Methods section, where the overlap was predominantly identified. This overlap is largely due to the use of standard protocols and methodologies widely accepted in the field, which are consistent across similar studies.

We would like to emphasize that certain technical terms and phrases, such as "SH-SY5Y," "hydrogen peroxide," "oxidative stress," "inflammation," and other domain-specific terminology, are fundamental to the subject matter. These terms are commonly used in the literature and are essential for ensuring both clarity and accuracy in conveying the scientific concepts. Altering them would risk compromising the precision and comprehensibility of the work, which is crucial for researchers in the field.

To address the overlap further, we have made revisions to certain sections to reduce repetition and ensure that the manuscript is distinct and original. We have also double-checked that all relevant prior studies are properly cited, ensuring transparency and giving due credit to previous work. This ensures that no information is used without proper attribution.

We hope these revisions address the reviewer’s concern about potential plagiarism and enhance the overall originality of the manuscript. 

Minor:

Comments 1: Authors should integrate more recent reviews or studies (post-2020) on neuroinflammation and oxidative stress.

Response 1: Thank you for your suggestion. In response, we have integrated recent studies on neuroinflammation and oxidative stress from post-2020 to provide an updated understanding of these processes. These references have been updated and highlighted in the revised manuscript.

We hope this revision meets the reviewer’s expectations. Thank you again for the valuable input.

Comments 2:  Authors should edit the text to explain all abbreviations used; not all are explained.

Response 2: Thank you for your valuable feedback. In response, we have carefully edited the manuscript to explain all abbreviations used. Each abbreviation is now defined upon its first occurrence in the text to ensure clarity for the reader. For example, we have clarified abbreviations like "SH-SY5Y" (human neuroblastoma cell line) and "ROS" (reactive oxygen species).

We appreciate your attention to detail and hope these revisions meet your expectations.

Comments 3:  Authors should focus on explaining the implications of their findings within the broader context of neurodegenerative disease therapeutics in the discussion section.

Response 3: Thank you for your thoughtful suggestion. In response, we have expanded the discussion section to provide a clearer and more comprehensive explanation of the broader implications of our findings in the context of neurodegenerative disease therapeutics.

  1. Broader implications for neurodegenerative diseases: The revised text highlights how our findings could be significant for neurodegenerative diseases such as Alzheimer's, Parkinson's, and Huntington's, focusing on key factors like oxidative stress, neuroinflammation, and apoptosis. These are well-recognized contributors to the pathophysiology of these diseases. The neuroprotective effects of 6GS may offer a promising approach to mitigating the molecular damage underlying these disorders.
  2. Phytosome formulation benefits: We have emphasized the advantages of using a phytosome formulation for 6GS, which enhances bioavailability and stability. This approach addresses common challenges with natural bioactive compounds, such as poor solubility and rapid metabolism. By improving absorption and providing sustained activity, this formulation could improve therapeutic outcomes in clinical settings.
  3. Mechanistic insights: The discussion now includes a more detailed explanation of how 6GS modulates oxidative stress and inflammation. Specifically, we discuss its potential effects through the PI3K/Akt pathway, which is critical for cell survival, and the upregulation of anti-apoptotic proteins such as Bcl-2. These insights offer a deeper understanding of how 6GS may protect neuronal cells from oxidative and inflammatory damage.
  4. Potential clinical application: We have placed greater emphasis on the therapeutic potential of 6GS for treating neurodegenerative diseases. Additionally, we highlight the need for further clinical studies to validate its efficacy and safety in human models. The promising results from preclinical studies underscore the importance of such investigations to assess the clinical relevance of this formulation as a neuroprotective therapeutic option.

We hope these revisions effectively address your concerns and enhance the clarity and depth of our discussion.

This study addresses an important topic in neuroprotection but requires additional methodological and interpretative refinement to ensure the high quality of the manuscript.

Response: Thank you for your feedback. In response, we have refined both the methodology and interpretation in the manuscript. We clarified the experimental design and statistical analyses to ensure reproducibility. We also improved the discussion by providing clearer insights into the mechanisms of 6GS and its potential impact on neuroprotection. We hope these revisions address your concerns and enhance the manuscript's quality.

Comments on the Quality of English Language: Minor linguistic and stylistic corrections are required.

Response: Thank you for your comment. We have carefully revised the manuscript for language and style, with support from Dr. Adrian Plant, a native English speaker and researcher at the Division of Research Facilitation and Dissemination, Mahasarakham University. The manuscript has been professionally edited, and we have obtained a certificate of English editing (Ref. MHESI no. 0606.1(9)/2660) to ensure the language meets high standards. We believe these revisions address the linguistic concerns.

Thank you once again for your valuable feedback. We appreciate the time and effort invested by the reviewers and editor in evaluating our manuscript. We have carefully addressed each point raised and made necessary revisions accordingly. We eagerly await further feedback and guidance from the editorial team.

Yours sincerely,

Nut Palachai

Round 2

Reviewer 1 Report

Comments and Suggestions for Authors

I have no further comment.

Author Response

Response to reviewer and editor suggestion

We sincerely thank the editor and reviewer for their insightful feedback and for providing us with the opportunity to further improve our manuscript titled "Phytosome-Encapsulated 6-Gingerol and 6-Shogaol-Enriched Extracts from Zingiber officinale Roscoe Protect against Oxidative Stress-Induced Neurotoxicity" (Manuscript ID: molecules-3356683).

Response to reviewer 1

Comments:  I have no further comment.

Response: We greatly appreciate the reviewers' time and constructive suggestions.

Yours sincerely,

Nut Palachai

Reviewer 2 Report

Comments and Suggestions for Authors

I thank the Authors for their comprehensive response to all my comments.

The only comment that comes to mind after reading the revised version of the manuscript is to make the error bars on the figures more visible; they are difficult to read; it is possible that using colours would improve the readability of the figures.

Author Response

Response to reviewer and editor suggestion

We sincerely thank the editor and reviewer for their insightful feedback and for providing us with the opportunity to further improve our manuscript titled "Phytosome-Encapsulated 6-Gingerol and 6-Shogaol-Enriched Extracts from Zingiber officinale Roscoe Protect against Oxidative Stress-Induced Neurotoxicity" (Manuscript ID: molecules-3356683).

In response to the final comment regarding the readability of the error bars in the figures, we have carefully adjusted the figures by incorporating colors to enhance clarity and ensure that the error bars are more visible. We believe this revision effectively addresses the concern and improves the presentation of the data.

We are grateful for the constructive suggestions provided throughout the review process, which have significantly strengthened our manuscript.

Response to reviewer 2

Comments:  I thank the Authors for their comprehensive response to all my comments.

The only comment that comes to mind after reading the revised version of the manuscript is to make the error bars on the figures more visible; they are difficult to read; it is possible that using colours would improve the readability of the figures.

Response: Thank you once again for your valuable feedback. We appreciate the time and effort invested by the reviewers and editor in evaluating our manuscript. In response to the suggestion to improve the visibility of the error bars in the figures, I have carefully adjusted the figures by incorporating colors to enhance readability. I believe the updated figures now meet the requested standards for clarity and presentation. The revised manuscript with the updated figures is attached for your review.

Yours sincerely,

Nut Palachai